

**Flexible vector-based spatial configurations in land models**
Shervan Gharari[1,*], Martyn P. Clark[1], Naoki Mizukami[2], Wouter J. M. Knoben[1], Jefferson S.
Wong[3], Alain Pietroniro[4]
1- University of Saskatchewan Coldwater Laboratory, Canmore, Alberta, Canada.
2- National Center for Atmospheric Research, Boulder, Colorado, USA.
3- Global Institute for Water Security (GIWS), Saskatoon, Saskatchewan, Canada.
4- Environment and Climate Change Canada (ECCC), Saskatoon, Saskatchewan, Canada.
*Corresponding author Shervan Gharari, shervan.gharari@usask.ca
**Abstract.** Land models are increasingly used in terrestrial hydrology due to their process-
oriented representation of water and energy fluxes. Land models can be set up at a range of
spatial configurations, often ranging from grid sizes of 0.02 to 2 degrees (approximately 2 to 200
km) and applied at sub-daily temporal resolutions for simulation of energy fluxes. A priori
specification of the grid size of the land models typically is derived from forcing resolutions,
modeling objectives, available geo-spatial data and computational resources. Typically, the
choice of model configuration and grid size is based on modeling convenience and is rarely
examined for adequate physical representation in the context of modeling. The variability of the
inputs and parameters, forcings, soil types, and vegetation covers, are masked or aggregated
based on the a priori chosen grid size. In this study, we propose an alternative to directly set up a
land model based on the concept of Group Response Unit (GRU). Each GRU is a unique
combination of land cover, soil type, and other desired geographical features that has
hydrological significance, such as elevation zone, slope, and aspect. Computational units are
defined as GRUs that are forced at a specific forcing resolution; therefore, each computational
unit has a unique combination of specific geo-spatial data and forcings. We set up the Variable
Infiltration Capacity (VIC) model, based on the GRU concept (VIC-GRU). Utilizing this model
setup and its advantages we try to answer the following questions: (1) how well a model
configuration simulates an output variable, such as streamflow, for range of computational units,



(2) how well a model configuration with fewer computational units, coarser forcing resolution
and less geo-spatial information, reproduces a model set up with more computational units, finer
forcing resolution and more geo-spatial information,  and finally (3) how uncertain the model
structure and parameters are for the land model. Our results, although case dependent, show that
the models may similarly reproduce output with a lower number of computational units in the
context of modeling (streamflow for example). Our results also show that a model configuration
with a lower number of computational units may reproduce the simulations from a model
configuration with more computational units. Similarly, this can assist faster parameter
identification and model diagnostic suites, such as sensitivity and uncertainty, on a less
computationally expensive model setup. Finally, we encourage the land model community to
adopt flexible approaches that will provide a better understanding of accuracy-performance
tradeoff in land models.
**1    Introduction**
Land models have evolved considerably over the past few decades. Initially, land models (or land-
surface models) were developed to provide the lower boundary conditions for atmospheric models
(Manabe, 1969). Since then land models have increased in complexity, and they now include a
variety of hydrological, biogeophysical, and biogeochemical processes (Pitman, 2003). Including
this broad suite of terrestrial processes makes land models suitable to simulate energy and water
fluxes and carbon and nitrogen cycles.
Despite the recent advancements in process representation in land models, there is currently
limited understanding of the appropriate spatial complexity that is justified based on the available
data and the purpose of the modelling exercise (Hrachowitz and Clark, 2017). The increase of
computational power, along with the existence of more accurate digital elevation models and land
cover maps, encourage modellers to configure their models at the finest spatial resolution possible.
Such hyper-resolution implementation of land models (Wood et al., 2011) can provide detailed
simulations at spatial scales as small as $1\text{-km}_2$ grid over large geographical domains (e.g., Maxwell
et al., 2015). However, the computational expense for hyper-resolution models could potentially
be reduced using more creative spatial discretization strategies (Clark et al., 2017).



It is common to adopt concepts of hydrological similarity to reduce computational costs. In this
approach, spatial units are defined based on similarity in geospatial data, under the assumption that
processes, and therefore parameters, are similar for areas within a spatial unit (e.g., Vivoni et al.,
2004). Hydrological Response Units (HRUs) are perhaps the most well-known technique to group
geospatial attributes in hydrological models. HRUs can be built based on various geospatial
characteristics; for example, Kirkby and Weyman 1974, Knudsen and Refsgaard (1986), Flügel
(1995), Winter (2001), and Savenije (2010) all have proposed to use geospatial indices to discretize
a catchment into hydrological units with distinct hydrological behaviour. HRUs can be built based
on soil type such as proposed by Kim and van de Giessen (2004). HRUs can also be built based
on fieldwork and expert knowledge (Naef et al., 2002, Uhlenbrook 2001), although the spatial
domain of such classification will be limited to the catchment of interest and the spatial extent of
the field measurements. HRUs are often constructed by GIS-based overlaying of various maps of
different characteristics and can have various shapes such as for non-regular (sub-basins), grid,
hexagon, or triangulated irregular network also known as TIN (Beven 2001, Marsh et al., 2012,
Oliviera et al., 2006, Pietroniro et al., 2007). Similar approaches are used in land models.
Traditionally land models use the tiling scheme where a grid box is subdivided into several tiles
of unique land cover, each described as a percentage of the grid (Koster and Suarez, 1992). Land
models are also beginning to adopt concepts of hydrological similarity (e.g., Newman et al., 2014;
Chaney et al., 2018).
A long-standing challenge is understanding the impact of grid size on model simulations (Wood
et al., 1988). The effect of model grid size can have a significant impact on model simulation
across scale especially if the model parameters are linked to characteristics which are averaged out
across scale (Bloschl et al., 1995). Shrestha et al. (2015) have investigated the performance of
CLM v4.0 coupled with ParFlow across various grid sizes. They concluded the grid size changes
of more than 100 meters can significantly affect the sensible heat and latent heat fluxes as well as
soil moisture. Also using CLM, Singh et al. (2015) demonstrated that topography has a substantial
impact on model simulations at the hillslope scale (~100 meters), as aggregating the topographical
data changes the runoff generation mechanisms. This is understandable as the CLM is based on
topographical wetness index (Beven and Kirkby 1979, Niu et al., 2005). However, Melsen et al.
(2016) evaluated the transferability of parameters sets across the temporal and spatial resolutions
for the Variable Infiltration Capacity (VIC) model implemented in an Alpine region. They



concluded that parameter sets are more transferable across various grid sizes in comparison with
parameter transferability across different temporal resolutions. Haddeland et al. (2002) showed
that the transpiration from the VIC model highly depends on grid resolution. It remains debatable
how model parameters and performance can vary across various grid resolutions (Liang et al.,
2004; Troy et al., 2008; Samaniego et al., 2017).
The representation of spatial heterogeneity is an ongoing debate in the land modelling community
(Clark et al., 2015). The key issue is to define which processes are represented explicitly and which
processes are parameterized. The effect of spatial scale on emergent behaviour has been studied
for catchment scale models – the concepts of Representative Elementary Areas (REA), or
Representative Elementary Watersheds (REW), were introduced to study the effect of spatial
aggregation on system-scale emergent behaviour (Wood et al., 1995, Reggiani et al., 1999). The
effect of scale on model simulations is not well explored for land models. More work is needed to
understand the extent to which the heterogeneity of process representations is sufficient for the
purpose of a given modelling application, and the extent to which the existing data can support the
model configurations (Wood et al., 2011, Beven et al., 2015).
In addition to the choice on model's spatial configurations, more work is needed to define the
appropriate structure of land models. While many studies in hydrology have evaluated how model
structure affects the smaller scale watershed response (Son and Sivapalan 2007, Clark et al., 2008,
Fenicia et al., 2011, Shafii et al., 2017), this issue has received limited attention in the land
modelling community (Desborough, 1999). Only recently, a few land models enable changing the
process formulations within a limited range of model structural assumptions (Noah-MP, Niu et al.,
2011, SUMMA, Clark et al., 2015) We explore effects of different choices of runoff generation
process representation in the model.
In this study, we configure the Variable Infiltration Capacity (VIC) model in a flexible vector-
based framework to understand how model simulations depend on the spatial configuration. The
remainder of this paper is organized as follows: In Section 2, we present the VIC model, its vector-
based implementation, and its coupling to the mizuRoute routing model. In Section 3 we describe
the design of the experiments. In Section 4 we describe the results of the experiments. Section 5





discusses the implication of spatial discretization strategies on large-scale land model applications.
The paper ends in Section 6 with conclusions of this study and implications for future work.

## 2    Land model and the routing model

*2.1    The Variable Infiltration Capacity (VIC) model*

The VIC model was developed as a simple land surface/hydrological model (Liang et al. 1994)
that has received applications worldwide (Melsen et al., 2016). In this study we use classic VIC
version 5 (VIC-5, Hamman et al., 2018). The key features of VIC are: (i) traditionally, the VIC
model (version 4 and earlier) simulates sub-daily energy variables with daily forcing of minimum
and maximum temperature, precipitation and wind speed. This enables the VIC model to be easily
forced with hydrological available data sets worldwide while being able to solve the energy fluxes
over sub-daily time periods. (ii) The VIC model combines sub-grid probability distributions to
simulate surface hydrology such as variable infiltration capacity formulation (Zhao, 1982) with
bio-physical formulations for transpiration (Jarvis et al., 1976).
The VIC model uses three soil layers to represent the subsurface. While each soil layer can have
various physical soil parameters (e.g., saturated hydraulic conductivity, bulk density), each layer
is assumed to be uniform across the entire grid regardless of the vegetation type variability in that
grid. The VIC model assumes a tile vegetation implementation within each grid similar to the
mosaic approach of Koster and Suarez (1992). To account for spatial variability in vegetation, the
VIC model allows for root depths to be adjusted for every vegetation type. The vegetation
parameters (e.g., stomatal resistance, LAI, albedo) are fixed for every land cover. The VIC model
can account for different elevation zones to account for temperature lapse rate given elevation
difference in a grid cell, and also for the distribution of precipitation over various elevation zones.

*2.2    The VIC-GRU implementation: a vector-based configuration for land models*

The VIC model is typically applied at regular grid. Figure-1a illustrates the typical VIC
configuration – here the modeler selects a cell size, and then the soil, vegetation and forcing files
are all aggregated or disaggregated to the target cell size. Original data resolution and spatial
distribution of soil, land cover and forcing data are lost. In this study, we configure the VIC model





using non-regular shapes, Grouped response Units (GRUs, Kouwen et al., 1993), depending on
the soil, vegetation, and topography. The GRUs hence describe unique characteristics of soil,
vegetation type, elevation, slope and aspect. Figure-1b presents an example of irregular GRUs
created through spatial intersections of the land use and soil types. These GRUs then can be forced
at the original resolution of forcing, or upscaled or downscaled values. Computational units can
then be constructed that intersect the GRUs with the forcing grid. Therefore, each computational
unit has unique geospatial data such as soil, vegetation, slope and aspect and is forced with a unique
forcing (a specific GRU forced with unique forcing).
The benefits of vector-based implementation of the VIC model based on the concept of GRU can
be summarized as follows:
**1- No grid and no assumption on grid size;** *Model resolution loses it meaning.* In
traditional VIC implementation, the modeler selects a grid resolution (which is often a regular
latitude/longitude grid). The soil parameters and forcing data from any resolution must be
aggregated, disaggregated, resampled or interpolated for every grid size. The land cover data is
only considered as a percentage for every grid and spatial location of the land cover is lost.
However in the VIC-GRU setup these decisions are only based on the input and forcing data that
are chosen to be used in the modeling practice and no upscaling or downscaling to grid size is
needed.
**2- The GRUs at the resolution of the forcing data logically represent the heterogeneity**
**of the input data (meteorological forcing and geospatial information).** A higher number of
computational units than the proposed setup will arguably provide an unnecessary computational
burden due to identical forcing data and geospatial information.
**3- Direct simplification of geospatial data.** The vector-based implementation makes the
direct aggregation of GRUs based on merging the geospatial data. It is easier to aggregate similar
soil types or similar forested areas into a unified GRU with basic GIS function (dissolving for
example) than this would be if all data had to be converted to a uniform grid first.
**4- Direct specification of physical parameters.** As each of the GRUs have specific type of
land cover, soil type and other physical characteristics, it is straightforward to specify parameter



values based on look up tables (i.e., no averaging, upscaling or smearing is needed). This is
favorable because the modeler does not need to make decisions about methods used for upscaling
of geophysical data at the grid level.

**5- The ability to compare and constrain the parameter values for GRUs and their**

**simulations.** The impact of land cover, soil type and elevation zone can be evaluated separately.
For example, the GRU concept makes it easier to test if forested areas generate less surface runoff
than grasslands. The new implementation of VIC simplifies using knowledge of geospatial
properties (e.g., soils data) and hydrological processes (e.g., expected fluxes for specific GRUs) to
constrain model parameter values. Similarly, the GRU concepts simplify regularization across
large geographical domains.

**6- Avoid unrealistic combinations of land cover, soil and elevation zone.** Unlike the

traditional VIC configuration, the proposed VIV-GRU approach avoids unrealistic configuration
of land cover, soil and elevation zones. An example is presented in Figure-2. This setup is with
two elevation zones partitioned at the tree line and two land cover types, forest below tree line and
bare soil above the tree line. The traditional VIC configuration assumes four different
combinations, including the unrealistic case of forest above the tree line. This issue is avoided in
vector-based setup of VIC-GRU as the set up will only include two GRUs with forest for lower
elevation and with bare soil for higher elevation.

**7- Possibility to incorporate additional data.** If needed, additional data such as slope and

aspect can be incorporated into the GRUs, accounting for changes in shortwave radiation or lapse
rates for temperature. These additional controls can be implemented outside of the model in the
forcing files. GRUs can be built also based on variation of leaf area index (LAI) giving an
additional layer of information in addition to the land cover type.

**8- Easier comparison of model simulations and in situ point-scale observation and**

**visualization:** The GRU implementation makes it easier to compare the point measurement to
model simulation as the model simulations preserve extent of geospatial features. The GRU
implementation also simplifies the comparison across GRUs; this comparison is very difficult in
the typical VIC implementation because of the need to upscale geophysical information to the grid
scale.





**9- Modular and controlled selection of models:** The GRU implementation identifies the
characteristics and spatial boundary of geospatial domains. A model might not be suitable for
processes of some of the geospatial domains. Alternatively, processes of a GRU that is beyond the
capacity of one model can be replaced with an alternative model. For example glaciers, can be
replaced with more suitable models while the configuration and forcings remain identical.
Consequently, the effect of features such as glacier can be better studied at larger scale
hydrological cycle as more expert models can be applied to glacier while the rest of the GRUs can
be simulated with a model that includes general processes.
*2.3    Structural changes in VIC-GRU*
We implemented several changes to the VIC process equations:
1- The VIC model uses the ARNO formulation, or its Nijssen representation, to represent
baseflow (Todini et al., 1996, Nijssen et al., 2001). In this study we simplify the VIC baseflow
formulation to a linear reservoir with one parameter, $K_{slow}$.
2- Preferential flow pathways are added to the VIC model by partitioning the runoff (fast
reacting component of the VIC model) into (1) an effective surface flow component; and (2)
recharge to the baseflow reservoir (interpreted as macropore flow). This partitioning is
parameterized based on the macropore fraction (for further reading on the implementation refer to
Gharari et al., 2019).
3- It is assumed that vegetation roots are restricted to the first two layers of the soil. This is
due to the simplification of the VIC baseflow formulation.
*2.4    mizuRoute, a vector-based routing scheme*
In this study, we use the vector-based routing model mizuRoute (Mizukami et al., 2016). Vector-
based routing models can be configured for separate computational units than the land model (e.g.,
configuring routing models using sub-basins derived from existing hydrologically conditioned
DEMs such as Hydrosheds, Lehner et al., 2006, or Merit Hydro, Yamazaki et al., 2019). This
removes the dependency of the routing to the grid size or GRUs configurations, and eliminates the
decisions that are often made to represent routing-related parameters at grid scale. Therefore we



can ensure that two model configurations with different geospatial configurations are routed using
the same routing configuration. The intersection between the computational units in the land model
and the sub-basins in the routing model defines the contribution of each computational units in the
land model to each river segment.

## 3   Data and methods

### 3.1   *Experimental design*

In this study, we configure the VIC model in a flexible vector-based framework to understand how
model simulations depend on the spatial configuration. We consider four different methods to
discretize the landscape for seven different spatial forcing grids (see Table 1). The landscape
discretization methods include (1) simplified land cover and soils; (2) full detail for land cover and
soils; (3) full detail for land cover and soils, including elevation zones; and (4) full detail for land
cover and soils, including elevation zones and slope and aspect. The different spatial forcing grids
are 4-km, $0.0625_\circ$, $0.125_\circ$, $0.25_\circ$, $0.5_\circ$, $1_\circ$, and $2_\circ$. This design enables us to separate our method to
discretize the landscape from the spatial resolution of the forcing data.
Experiments are performed for the Bow River at Banff with a basin area of approximately 2210
$km_2$. The Bow River is located in the Canadian Rockies in the headwaters of the Saskatchewan
River Basin. Most of the Bow River streamflow is due to snow melt (Nivo-glacial regime). The
average basin elevation is 2130 m ranging from 3420 m at the peak top to 1380 m above mean sea
level at the outlet (town of Banff). The basin annual precipitation is approximately 1000 mm with
range of 500 mm for the Bow Valley up to 2000 mm for the mountain peaks. The predominant
land cover is conifer forest in the Bow Valley and bare soil and rocks for mountain peaks above
the tree line.
We design three experiments:

### 3.1.1   Experiment-1: How does the spatial configuration affect model performance?

As the first experiment, we focus on how well the various configurations simulate observed
streamflow at the  Bow River at Banff. We calibrate the parameters for the different configurations
in Table 1. Model calibration is accomplished using the Genetic Algorithm implemented in the



OSTRCIH framework (Mattot, 2005; Yoon and Shoemaker, 2001), maximizing the Nash-Sutcliffe
Efficiency ($E_{NS}$, Nash and Sutcliffe 1970) using a total budget of 1000 model evaluations given
the available resources limited by the most computationally expensive model (Case-4-4km).

### 3.1.2   Experiment-2: How well do calibrated parameter sets transfer across different model configurations?

As the second experiment, we focus on how various configurations can reproduce the result from
the configuration with highest computational units for   a given parameter set. In other words, this
experiment  evaluates  accuracy-efficiency  tradeoffs – i.e.,  the  extent  to  which  spatial
simplifications  affect  model  performance  under  the  assumption  that  similar  GRUs  possess
identical parameters across various configurations. This is important as it enables modelers to
understand efficiency-accuracy tradeoffs, given the available data and the purpose of the modelling
application. This experiment is based on perfect model experiments using the model with the
highest computational unit as synthetic case (Case-4-4km). Synthetic streamflow for every river
segment is generated using a calibrated parameter set for Case-4-4km-4km. The models with lower
number of computational units are then simulated using the exact same parameter set used for
generating the synthetic streamflow. The differences in streamflow simulation, quantified using
$E_{NS}$, provide an understanding of how the simulations deteriorate when the spatial and forcing
heterogeneities are masked or up-scaled. This also will bring an understanding on how sensitive
the changes are along the river network and at the gauge location at which the models are calibrated
against the observed streamflow data. Similarly, we compare the spatial patterns of snow water
equivalent for the different spatial configurations.

### 3.1.3   Experiment 3: How do different model structures affect model performance?

As the third experiment, we focus on the effect of model structure on the performance metric ($E_{NS}$).
This experiment, although not directly linked to the exploration of spatial configuration, is
designed to investigate the effect of model structure changes on model performance which may
affect our perception of parameter allocation across the GRUs (non-uniqueness of models,
processes and parameter values)... For Case-2-4km, we calibrate the model with macropores
activated and micropore deactivated. We call this model Case-2-4km-macro. We compare the



general model behavior looking into surface runoff and base-flow proportions of the streamflow
for GRUs for the two model setups, Case-2-4km and Case-2-4km-macro.
*3.2   Geospatial data and meteorological forcing*
The inputs and forcing we used to set up the models are as follows:
1- Land cover: We used the land cover map NALCM-2005 v2 that is produced by CEC
(Latifovic et al., 2004). NALCM-2005 v2 includes 19 different classes. The land cover map is used
to set up the vegetation file and vegetation library (look up table) for the VIC model (Nijssen et
al., 2001).
2- Soil texture: We used the Harmonized World Soil Data, HWSD (Fischer et al., 2008). For
each polygon of the world harmonized soil we use the highest proportion of soil type. The HWSD
provide the information for two soil layers, in this study we base our analyses on the lower soil
layer reported in HWSD to define the soil characteristics needed for the VIC soil file.
3- Digital Elevation Model: in this study we make use of existing hydrologically conditioned
digital elevation models to (1) derive the river network topology for the vector-based routing,
mizuRoute and (2) to derive the slope, aspect and elevation zones which are used to estimate the
forcing variables. For the first purpose we use hydrologically condition DEM of HydroSHED with
resolution of 3 arc-second, approximately 90 meters; for the second purpose we use HydroSHED
15 arc-second DEM (approximately 500 meters).
4- Meteorological forcing: we used the WRF data set with the temporal resolution of 1 hour
and spatial resolution of 4 km (Rasmussen and Liu, 2017). For upscaling the WRF input forcing,
we use the CANDEX package (DOI: 10.5281/zenodo.2628351) to map the 7 forcing variables to
various resolutions (1/16˚, 1/8˚, 1/4˚, 1/2˚, 1˚ and 2˚ from the original resolution of 4 km). We used
the required variables from the WRF data set namely, total precipitation, temperature, short and
long wave radiation at the ground surface, V, U components of wind speed and water vapor mixing
ratio.
The shortwave radiation is rescaled based on the slope and aspect of the respective GRUs (refer to
Appendix-A for more details). In this study we differentiated four aspects and five slope classes.



The temperature at 2 meters are adjusted using the environmental lapse rate for temperature of 6.5
km per 1000 meters. The assumed lapse rate aligns with earlier findings from the region of study
(Pigeon and Jiskoot, 2008).

### 3.3   Observed data for model calibration

The daily streamflow is extracted from the HYDAT (WSC, Water Survey Canada) for Bow at
Banff with gauges ID of 05BB001. This data is used for parameter calibration/identification of
VIC-GRU parameter values.

### 3.4   Model parameters

#### 3.4.1   VIC-GRU parameters

In the experiments for this study, we calibrate a subset of VIC parameters namely $b_{inf}$, $E_{exp}$, $K_{sat}$,
$d_{2,forested}$, $d_{2,non-forested}$ and $K_{slow}$ and $D_{macro-fract}$ (names are mentioned in Table-2). We make sure that
the $d_{2,\ forested}$ is larger than the $d_{2,non-forested}$ as the root depth are deeper for forested regions
(constraining relative parameters).

#### 3.4.2   MizuRoute parameters:

Impulse Response Function (IRF) routing method (Mizukami et al., 2016) is used for this study.
IRF, which is derived based on diffusive wave equation, includes two parameters – wave velocity
and diffusivity. The parameters for the routing scheme and river network topology for the
mizuRoute is identical for all the configurations and experiments. The river network topology,
assuming 100 km2 starting threshold for the sub-basin size, is based on a 92-segment river network
depicted in Figure-3d. The diffusive wave parameters are set to 1 m/s and 1000 m2/s respectively
and remain identical for all the river segments.

## 4   Results

### 4.1   Experiment-1

The various model configurations are compared with respect to the Nash-Sutcliffe performance
metric ($E_{NS}$). Results show that all the models, including the ones that are configured with coarser





resolution forcings, can simulate streamflow with $E_{NS}$ as high as 0.70 (Table-3). These results
indicate that the coarse resolution forcing input and lower computational units are able to yield
equivalent $E_{NS}$ of 0.7 and higher.
Although the performance metric of the various configurations, it is noteworthy to mention that
the configuration of Case-4-0.5° has higher $E_{NS}$ value compared to the cases with highest
computational units, Case-4-4km for example. This might be due to various reasons including: (1)
compensation of forcing aggregation on possible forcing bias at finer resolution; (2) compensation
of forcing aggregation on model states and fluxes and possible adjustment for model structural
inadequacy and hence directing the optimization algorithm to different possible solutions across
configurations.
The model simulations, with $E_{NS}$ higher than 0.7 for example, have very different soil parameters
configuration. As an example, saturated hydraulic conductivity, $K_{sat}$, and slope of water retention
curve, $E_{exp}$, can have very different combinations of values within the specified ranges for the
parameters. Figure-4 illustrates the possible combinations of $K_{sat}$ and $E_{exp}$ with performance higher
than $E_{NS}$ greater than 0.7 for Case-2-4km. The result indicates the two parameters that are often
fixed or a priori allocated based on look up tables can exhibit significant uncertainty and non-
identifiability. Moreover, calibrating the VIC model using a sum-of-squared objective function at
the basin outlet does not constrain the VIC soil parameters.
*4.2   Experiment-2*
The second experiment compares the performance of a parameter set with $E_{NS}$ of above 0.7 from
the Case-4-4km across the configurations with degraded geophysical information and aggregated
spatial information. Figure-5 shows the evaluation metric, $E_{NS}$, for the streamflow of every river
segment across the domain in comparison with the synthetic case (Case-4-4km). From Figure-5,
it is clear that the $E_{NS}$ is less sensitive for river segments with larger upstream area (read more
downstream). This result has two major interpretations (*i*) the parameter transferability across
various configuration is dependent on the sensitivity of simulation at the scale of interest and (*ii*)
often inferred parameters at larger scale may not guarantee good performing parameters at the
smaller scales.





Figure-6 shows the performance of the streamflow across various configurations for the most
downstream river segment (the gauged river segment which is often used for parameter inference
through calibration). Figure 6 illustrates that most of the configurations have similar scaled $E_{NS}$ at
the basin outlet. This analysis can be repeated for different parameter sets, e.g., poorly performing
parameter sets or randomly selected parameter sets, to better understand accuracy-efficiency
tradeoffs. Such analyses can provide insights on the appropriate model configurations for different
applications. As an example, if model configurations of different complexity are known to show
similar performance for a given parameter set, uncertainty and sensitivity analysis can be done
initially on the models with fewer computational units and the results of the analysis can be applied
to models with a higher number of computational units. This is however under assumption that
parameters are transferable based on the concept of GRU.
To understand the spatial patterns of model simulations for all the configurations, we evaluate the
distribution of the snow water equivalent, SWE, for the computational units on 5th of May 2004
(Figure-7). In general, the SWE follows the forcing resolution and its aggregation. Although
coarser forcing resolutions results in coarser SWE simulation, the geospatial details such as
elevation zones and slope and aspects result in more realistic representation of SWE as the snow
layer is thinner for south facing slopes where more melt can be expected to occur, and thicker for
higher elevation zones (compare SWE simulations for Case-4-2° and Case-3-2° in Figure-7) which
is consistent with higher precipitation volumes and slower melt at higher elevation. Another
observation from Figure-7 is the unrealistic distribution of SWE for configurations without
elevation zones (Case-2 and Case-1). The lack of elevation zones results in both valley bottom and
mountain tops to be forced with the same temperature. Snow is more durable in the forested areas
as the result of model formulation, which are at lower elevation, while SWE is less for higher
mountains, which is unrealistic.
We compared the maximum snow water equivalent across different configurations for a
computational unit located in the Bow Valley Bottom (an arbitrary location of -116.134°W and
51.382°E) for the year 2004. Figure-8 illustrated the maximum snow water equivalent for the
period of simulation. The result indicates that the SWE is higher for configurations with coarser
forcing resolutions (almost double). This is due to the reduced temperature as a result of masking
warmer valley bottom by cooler and higher forcing grids over the Rockies.





*4.3   Experiment-3*

The calibration of model Case-2-4km and Case-2-4km-macro result in similar $E_{NS}$ values of 0.78 and 0.75, indicating that both models are able to reproduce the observed streamflow to a similar extent although they are structurally different (the slow reservoir is recharged through only micropore and only macropore water movement in Case-2-4km and Case-2-4km-macro respectively). Figure-9 shows the streamflow hydrographs for the best performing parameter sets. . Observed Bow River at Banff has a minimum streamflow of approximately 15 cubic meter per second during snow accumulation months. This flow may be the result of regulation, return flow from human activities or unaccounted processes such as groundwater flow which are rather difficult for the linear reservoir of baseflow to simulate. Results (not shown) also indicate that both models structures generate 4 to 5 times more baseflow than surface runoff. This might be very intuitive as the model structure and parameters only have one processes, the slow reacting component, to simulate the long memory of this Nivo-glacial system and its annual cycle. Even though the two models are structurally different, both produce flow volumes through the surface and baseflow pathways that are consistent with streamflow observation. Similar to the uncertainty of model parameters, this result also shows the uncertainty of model structure and the fact that inclusion or exclusion of macropore water movement in the region of study and the context of modeling, may not change the overall results and similarly that these processes, micropore or macropore flow, cannot be inferred from the streamflow observation only.

## 5   Discussion

In this study, we proposed a vector-based configuration for land models and applied this setup to the VIC model. We used a vector-based routing scheme, mizuRoute, which was forced using output from the land model (one-way coupling). We term this new modelling approach VIC-GRU. Unlike the grid-based approach, there is no upscaling of land cover percentage or soil characteristics to a new grid size. This enables us to separate the effects of changes in forcing from changes in the spatial configurations. The vector-based setup also provides us with more flexibility in comparing the model simulations across GRUs, and also comparing model simulations with point measurements, such as snow water equivalent.


Our results illustrate that the VIC-GRU approach generates similar large-scale simulations of
streamflow across the various spatial configurations when VIC-GRU is calibrated by maximizing
the Nash-Sutcliffe score at the basin outlet. Similarly, we have shown that the VIC soil parameters
can be very different when calibrated using different spatial configurations and that parameter
transferability to different forcing resolutions and model setups is limited. This uncertainty is not
often evaluated or reported for land models (Demaria et al., 2007) or is ignored by tying
parameters, linking specific hydraulic conductivity to the slope of water retention curve, for
example, so that the possible combination of them are reduced.
Land models are often applied at large spatial scales. The results clearly show that the deviation
of streamflow is much lower in river segments with larger upstream area (Figure 5 and 6). It is
often the case that the model parameters are inferred based on calibration on the streamflow at the
basin outlet or over a large contributing area. We argue that this may not be a valid strategy for
process understanding at the GRU scale, given the large uncertainty exhibited by the parameters.
Therefore, hyper-resolution modeling efforts, Wood et al. 2011, may suffer from poor process
representation and parameter identification at the scale of interest (Beven et al., 2015). What is
needed instead of efficiency metrics that aggregate model behavior across both space (e.g. at the
outlet of the larger catchment) and time (e.g. expressing the mismatch between observations and
simulations across the entire observation period as a single number), is diagnostic evaluation of
the model's process fidelity at the scale at which simulations are generated (e.g. Gupta et al., 2008;
Clark et al., 2016).
We have shown that changes in model structure can result in identical performance for system-
scale evaluation metrics (in this case, the Nash Sutcliffe Efficiency). We have changed the land
model structure by replacing the micropore with macropore water movement to the slow reservoir.
Similar to the parameter uncertainty, this indicates that lack or inclusion of macropore processes
at the GRU scale does not have any notable effect on the efficiency score of the model simulation
of streamflow at the outlet of the basin, even though the process simulations at the GRUs are
different. Alternatively, this also shows that the micropore and macropore processes and their
parameters may not be identifiable through calibration on the observed streamflow, which supports
the argument against assuming that fine-scale parameters and processes can be inferred from large-
scale observations. Although in this study we only focus on the processes and parameters that are



often used to calibrate for the VIC model such as subsurface processes, it is possible to repeat the
same analysis on wider range of processes such as snow processes or routing parameters.
It is often computationally expensive to evaluate the uncertainty and sensitivity of land models.
Following the results presented in Figure-6, one can assume a configuration with fewer
computational units can be a surrogate for a model with more computational units, under the
condition that both models are known to behave similarly for a given parameter set. The calibration
can be done on the model configuration with less computational unit and the parameters can be
transferred directly to the model with more computational units, or can be used as an initial point
for optimization algorithm to speed up the calibration process. Similarly the sensitivity analyses
can be done primarily on the model with less computational units.
One might argue that the spatial discretization is important for realism of model fluxes and states.
Moving to significantly high number of GRUs may result in computational units that are similar
in their forcing and spatial variability. Based on the result of this study for snow water equivalent
(Figure-8), we can argue that the snow patterns are fairly similar for the configurations that have
elevation zones and finer resolution of forcing (case3 and 4 and forcing resolution less than 0.125
degree). It can be further explored if the model simulation at finer resolutions can be approximated
by interpolating result of a model with coarser resolution ($m(\bar{x}|\theta) \sim \overline{m(x|\theta)}$, in which $m$ is the
model, $x$ is forcing and $\boldsymbol{\theta}$ is the parameter set).
In this study and following the concept of GRUs, Grouped Response Units, we assumed the
physical characteristics of soil and vegetation are identical for a given GRU across various model
configurations. Techniques such as multiscale parameter regionalization (MPR, Samaniego et al.,
2010) can be used to scale parameter values for different model configurations. However, applying
these techniques, such as in this case that has significant parameter and process uncertainty and
significance accuracy-performance tradeoff, should be put through rigorous tests (Merz et al.,
2020, Liu et al., 2016).
Also, the degree of validity of the concept of GRU, hydrological similarity basd on physical
attribute similarities, is debatable. For example, at the catchment scale, Oudin et al. (2010) have
shown that the overlap between catchments with similar physiographic attributes and catchments
with similar model performance for a given parameter set is only 60%. Physiographic similarity



(in our case expressed through GRUs) does thus not necessarily imply similarity of hydrologic
behavior, even though this is the critical assumption underlying GRUs. Although the GRUs in this
study include slope and aspect, these characteristics were not translated into the model parameters
and was only used for forcing manipulation. The VIC parameters can be linked to many more
characteristics such as slope, height above nearest drainage (HAND, Renno et al., 2008), or
Topographical Wetness Index (Beven and Kirkby, 1979) as has been done by Mizukami et al.
(2017) and Cheney et al. (2018). However the functions that are used to linked the attribute to
model characteristics remains mostly assumptions rather than inference and reproducibility of
them are not very well explored (if possible).
In this study, the vector-based routing configuration does not include lakes and reservoirs. This is
often a neglected element of land modeling efforts and has only attracted limited attention
compared to the its impact on terrestrial water cycle (Haddeland et al., 2006, Yassin et al., 2018).
The presence of lakes and reservoirs and their interconnections reduces the, already limited, ability
of inference of land model parameter based on calibration on the observed streamflow because
streamflow variability is reduced.
Although not primary the result of this study, however, the Nivo-glacial regime of the Bow River
Basins is mostly dominated by snow melt that contributes to streamflow through baseflow (slow
component of the hydrograph). The high Nash-Sutcliffe Efficiency, $E_{NS}$, is partly due to the fact
that it is rather easy for the land model to capture the yearly cycle of the streamflow only with
snow processes while rapid subsurface water movement, such as macropore, are largely missing
in the land models but do not lead to notably increased efficiency scores when they are included
in the model structure. More caution is needed for use of the land model for flood forecasting
(Vionnet et al., 2019) for this region and all the Nivo-glacial river systems in western Canada,
McKenzie, Yukon and Colombia River Basins.
**6   Conclusions**
The vector-based setup for the land model can provide modelers with more flexibility, e.g. impact
of various forcing resolution or geospatial data representation, while avoiding decisions that are
often taken for model configuration at grid level. The conclusion and messages from this study
can be summarized as follows:





1) Regardless of observations at the scale of modeling, a model configuration with lower
computational units, coarser resolution and less geospatial information, can produce model
simulations with similar efficiency scores as configurations with higher computational
units. The choice of model set up should be tested within the context and purpose of
modeling for every different case.

2) The model with the highest number of computational units may not result in improved
performance and better spatial simulation, in terms of obtained efficiency scores, and
parameters can be transferred without substantial performance changes between certain
model setups. Less computationally expensive configurations can be used instead for
primary uncertainty and sensitivity analysis.

3) There is significant parameter and structural uncertainty associated with the VIC-GRU
model. This uncertainty creates challenges for the process and parameter inference using
calibration on streamflow. Any regionalization for parameters of the model should take
into account these significant uncertainties. Our results recommend caution and more
attention to the topic of parameter and process inference at finer modelling scales.

We also encourage the need for tools which can facilitate easier and more flexible set up of land
models that in turn can facilitate the above mentioned research questions.
***Acknowledgment.*** This research was undertaken thanks in part to funding from the Canada First
Research Excellence Fund.
***Data availability.*** All the data used in this study are available publicly (refer to references).
**7    Appendix**
*7.1    Appendix – A*
This appendix reflect on the methods and equations that have been used to calculate the ration of
the solar radiation to flat surface and a surface with slope and aspect.
**Declination angle:** declination angle can be calculated for each day of year and is the same for
the entire Earth based on (Ioan Sarbu, Calin Sebarchievici, in Solar Heating and Cooling Systems,
2017):





$$\delta = 23.45 \frac{\pi}{180} \sin\left[\frac{2\pi}{360}\frac{360}{365}(284 + N)\right]$$     (A-1)
In which N is the number of day in a year starting from beginning 1st of January.
**Hour angle:** is the angle expressed the solar hour. The reference of solar hour angle is solar noon
(hour angle is set to zero) when the sun is passing the meridian of the observer or when the solar
azimuth is 180. The hour angle can be calculated based on the:
$$\sin\omega = \frac{\sin\alpha - \sin\delta\sin\emptyset}{\cos\delta\cos\emptyset}$$     (A-2)
In which α, ϕ and δ are the altitude angle, latitude of the observer and inclination angle.
The solar noon is not exactly coinciding with12 am of the local time zone. However in this study
we assume the two property are coinciding. The sunset and sunrise hour can be calculated from:
$$\cos\omega_s = -\tan\emptyset\tan\delta$$     (A-3)
For beyond 66.55 degree if the value of the right hand side is above 1 then there is 24 hour of
daylight and if the right hand side is less than 1 the will be 24 hour of darkness.
The number of daylight hours that can be split before and after the solar noon equally can be
calculated based on (assuming 15 degree for every 1 hour):
$$n = \frac{2\omega_s}{15}\frac{180}{\pi}$$     (A-4)
**Altitude angle:** is the angle of sun with the observer. This angle is maximum at solar noon and 0
for subset and sunrise. The altitude angle can be calculated based on the:
$$\sin\alpha = \sin\delta\sin\emptyset + \cos\delta\cos\omega\cos\emptyset$$     (A-5)
For the solar noon when ω, hour angle, is zero the question simplifies to:
$$\sin\alpha = \sin\delta\sin\emptyset + \cos\delta\cos\emptyset = \cos(\emptyset - \delta) = \sin\left(\frac{\pi}{2} - \emptyset + \delta\right)$$     (A-6)
This result the altitude angle for the solar noon to be:





$\alpha = \frac{\pi}{2} - \emptyset + \delta$                                               (A-7)
**Solar Azimuth:** The solar azimuth angle, $\theta_{Sun}$ reflect on the angle of the sun on the sky from the
North with clockwise rule. The azimuth angle can be calculated as:
$\sin \theta_{Sun} = \frac{\sin \omega \cos \delta}{\cos \alpha}$                                  (A-8)
The solar azimuth angle for the solar noon is set to be 180 degree (calculated clockwise from north
direction).
The azimuth at the sunset and sunrise can be calculated using:
$\sin \theta_{Sun,rise} = -\sin \omega_s \cos \delta$                                      (A-9)
$\sin \theta_{Sun,set} = \sin \omega_s \cos \delta$                                        (A-10)
**Surface Azimuth (a.k.a. aspect):** The surface azimuth angle, $\theta_{Surface}$ reflect the direction of the
any tilted surface to the north direction. This azimuth is fixed for any point while the solar azimuth
changes over hours and seasons.
**Angle of incidence $\theta$:** this angle represent the angle between a sloped surface and the sun rays
that reaches this sloped surface. The model angle of the incidence for a slope surface β, and aspect
of $\theta_{Surface}$ over latitude of $\emptyset$ can be calculated as (Kalogirou, in Solar Energy Engineering, 2009,
in the reference formulation the Azimuth is from south which is corrected here for North):
$\cos \theta = \sin \delta \sin \emptyset \cos \beta + \sin \delta \cos \emptyset \sin \beta \cos \theta_{Surface} + \cos \delta \cos \emptyset \cos \beta \cos \omega -$
$\cos \delta \sin \emptyset \sin \beta \cos \theta_{Surface} \cos \omega - \cos \delta \sin \beta \sin \theta_{Surface} \sin \omega$         (A-11)
For the flat surface, both $\theta_{Surface}$ and β, is set to zero, the incident angle can be calculated for the
flat surface as
$\cos \theta_{flat} = \sin \delta \sin \emptyset + \cos \delta \cos \emptyset \cos \omega$                         (A-12)
In case where the angle of incident is larger than 90 degrees the surface shades itself.



**Amendment of short wave radiation based on slope and aspect.** In this study we correct the
WRF short wave radiation based on the surface slope and aspect. We first back calculated the
incoming short wave radiation by dividing the provided short wave radiation by the cosine of the
incident angle of the flat surface. Then we can calculate the solar radiation of the sloped surface
multiplying this value to the cosine of the incident angle of the slope surface. Basically this ratio
is:
$$R = \frac{\cos\theta}{\cos\theta_{flat}} \qquad\qquad\qquad\qquad \text{(A-13)}$$
The effect of the atmosphere is considered in the WRF product itself. However, and for incident
level close to 90 degrees the ratio, *R*, might be very high values which result in the surface
receiving unrealistically high value of radiation even higher than the solar constant, 1366 W/m2,
at the top of the atmosphere. For cases with cos values of incident angle lower than 0.05 we set
the ratio to 0 to avoid this unrealistic condition.

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

# 9    Tables
Table – 1 the number of computational units for the Bow River at Banff, given different spatial
discretization of land cover, soil type, elevation zones and slope and aspects forced with various
forcing resolutions.

| | Forcing resolution | Case 4<br>4 aspect groups;<br>5 slope groups;<br>19 classes of land cover;<br>500 meter elevation zones; | Case 3<br>no aspect groups;<br>no slope groups;<br>19 classes of land cover;<br>500 meter elevation zones; | Case 2<br>no aspect groups;<br>no slope groups;<br>19 classes of land cover;<br>no elevation zones; | Case 1<br>no aspect groups;<br>no slope groups,<br>3 classes of land cover,<br>one dominant soil type<br>no elevation zones; |
|---|---|---|---|---|---|
| Number of GRUs | -- | **582** | **65** | **56** | **3** |
| Number of Computational units (GRUs forced at various forcing resolutions) | 4km | 6631 | 1508 | 941 | 479 |
| | 0.0625 | 5224 | 1098 | 663 | 290 |
| | 0.125 | 3079 | 515 | 283 | 94 |
| | 0.25 | 2013 | 306 | 154 | 39 |
| | 0.5 | 1332 | 184 | 93 | 21 |
| | 1.0 | 917 | 116 | 56 | 12 |
| | 2.0 | 767 | 89 | 42 | 6 |









Table-2 the VIC-GRU model parameters that are subjected to perturbation for model calibration
for the designed experiments.

| Parameter symbol | Parameter name | Minimum value | Maximum value | Unit | Explanation |
|---|---|---|---|---|---|
| $b_{inf}$ | Variable infiltration parameter | 0.01 | 0.50 | [-] | |
| $E_{exp}$ | The slope of water retention curve | 3.00 | 12.00 | [-] | |
| $K_{sat}$ | Saturated hydraulic conductivity | 5.00 | 1000.00 | [mm/day] | Fixed at very low rate, 0.0001, for the macropore model in experiment 3 to diable micropore water movement to the slow reservoir. |
| $d_1$ | The depth of top soil layer | 0.2 | 0.2 | m | Fixed at 20 cm for both forested and non-forested GRUs |
| $d_{2,forested}$ | The depth of the second soil layer for forested GRUs | 0.2 | 2 | m | |
| $d_{2,non-forested}$ | The depth of the second soil layer for non-forested GRUs | 0.2 | $d_{2,forested}$ | m | The maximum is bounded by the $d_{2,forested}$ |
| $D_{root}$ | The distribution of root in the two soil layers. | 0.5 | 0.5 | | Fixed at 50% for the top and lower soil layers. |
| $K_{slow}$ | Slow reservoir coefficient | 0.001 | 0.9 | [1/day] | |
| $D_{macro-fract}$ | Macropore fraction | 0.0 | 1.0 | [-] | Fixed at 0.00 for experiment 1 and experiment 2, varying for experiment 3. |










Table-3 – The $E_{NS}$ for the different model configurations. Details on the geospatial cases are
provided in Table 1.

| Forcing resolution | Case 4 | Case 3 | Case 2 | Case 1 |
|---|---|---|---|---|
| 4km | 0.80 | 0.80 | 0.78 | 0.74 |
| 0.0625° | 0.80 | 0.80 | 0.78 | 0.77 |
| 0.125° | 0.80 | 0.80 | 0.76 | 0.73 |
| 0.25° | 0.82 | 0.81 | 0.76 | 0.76 |
| 0.5° | 0.84 | 0.84 | 0.76 | 0.75 |
| 1.0° | 0.82 | 0.81 | 0.78 | 0.78 |
| 2.0° | 0.78 | 0.78 | 0.72 | 0.76 |

















**10 Figures**

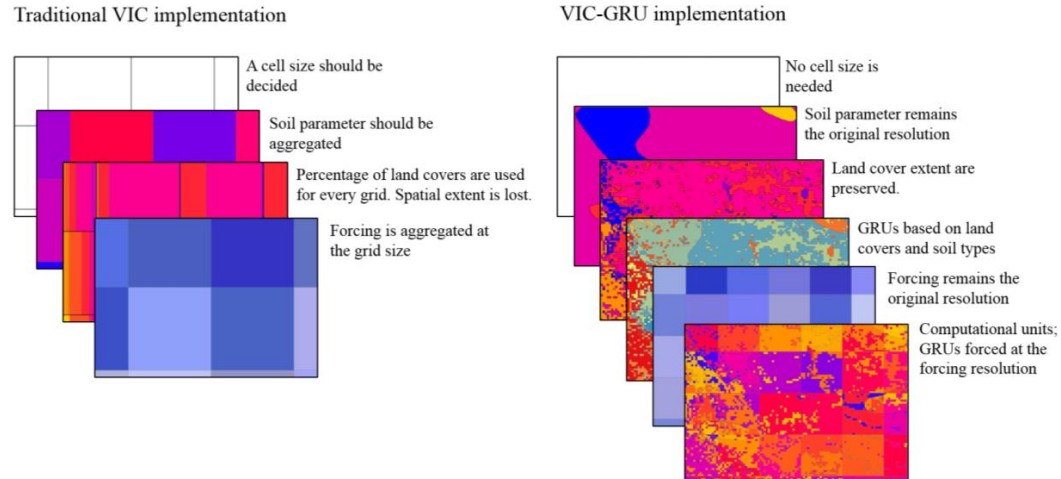


Figure-1- (a) Traditional VIC implementation and (b) new VIC implementation (VIC-GRU).

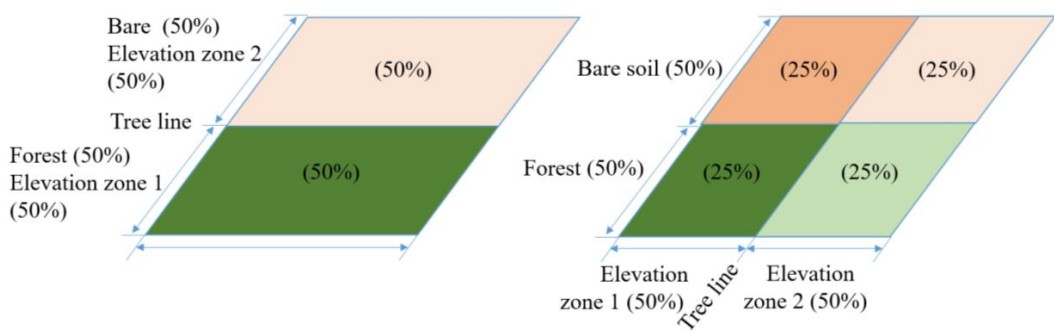


Figure-2 – (Left) the true configuration of a natural system with land cover consist of 50% Bare
soil and 50% forest within a grid located in two different elevation zones above and below the tree
line and (right) the traditional VIC configurations for the given system at the grid for the two
elevation zones and 2 land cover which results in unrealistic combination of forest cover above
the tree line and bare soil below the tree line.


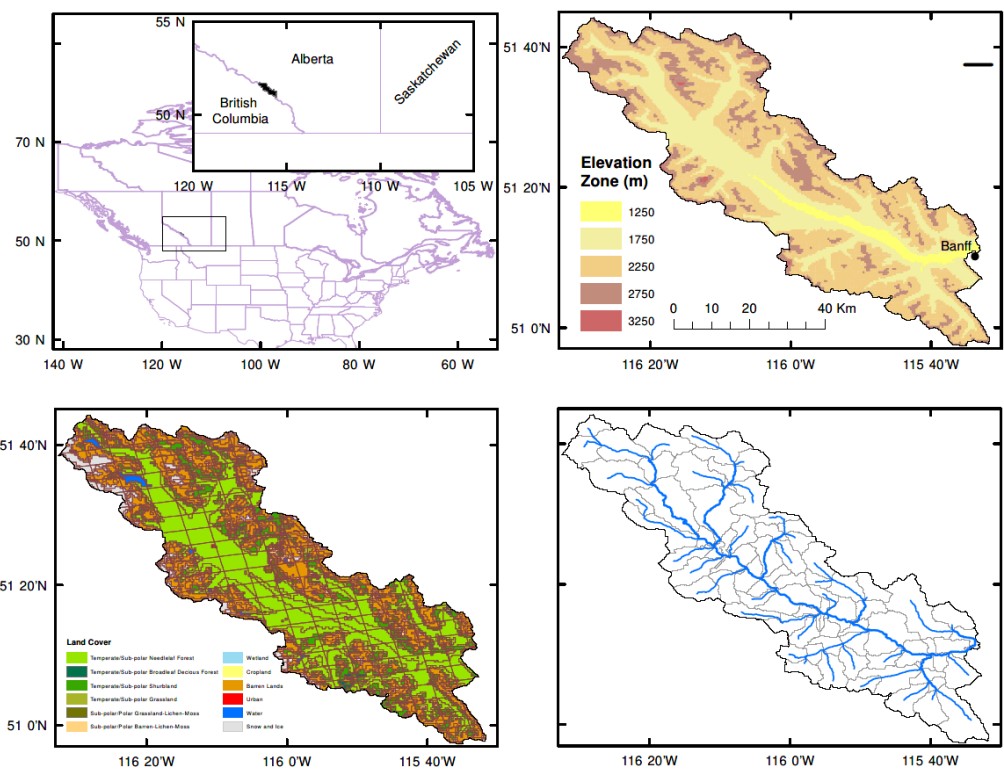

Figure – 3 (a) The location of the Bow River Basin to Banff (b) GRUs for Case-3 color-coded for
elevation zones, (c) computational units for the Case-3-4km (Case-3 forced at forcing of 4 km
resolutions) and (d) river network topology and associated sub-basins for the vector-based routing.





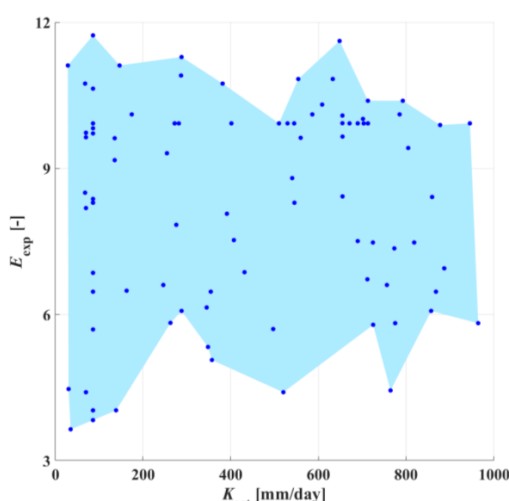


Figure-4 – The spread of two parameters, saturated hydraulic conductivity, $K_{sat}$, and slope of water
retention curve, $E_{exp}$, for the parameters sets that have performance metric, $E_{NS}$, of more than 0.7
for configuration Case-2-4km. The axis are set to the ranges of the parameters.




Figure 5 – Deviation of the simulated streamflow at river segments in comparison with the
synthetic case of GRUs forced at 4km, Case-4-4km, expressed in performance metric, $E_{NS}$.






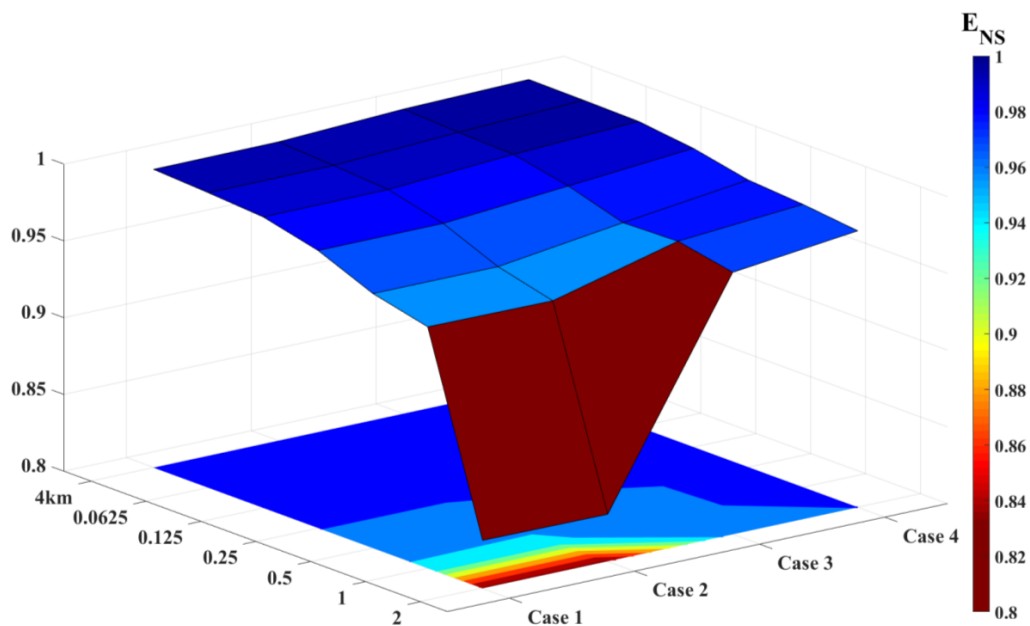


Figure -6- The relative performance of model simulation across various configurations with a
single parameter set.



Figure 7- Comparison of the snow water equivalent for 5th of May 2004 for various configurations.


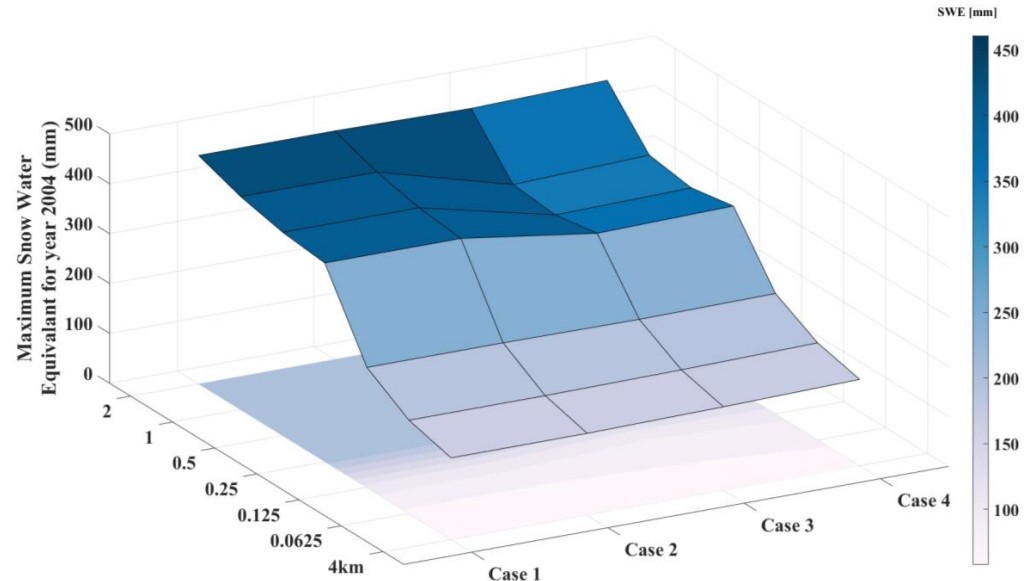


Figure-8- Maximum of snow water equivalent for an arbitrary location of -116.134˚W and
51.382˚E located in Bow Valley Bottom across various model configurations for the year 2004.

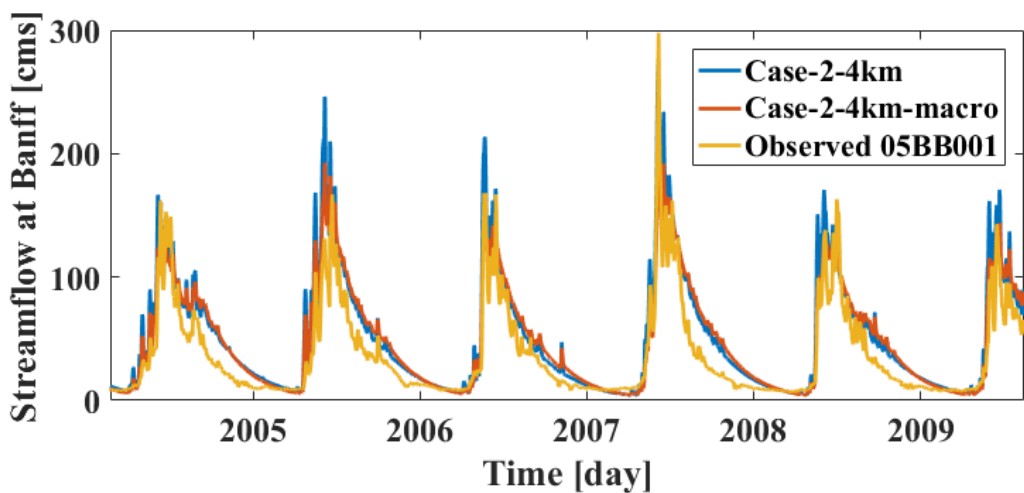


Figure-9 comparison between the streamflow observation for Bow at Banff (05BB001) and model
with only micropore flow to slow reservoir, Case-2-4km, and only macropore to the slow reservoir,
Case-2-4km-macro.




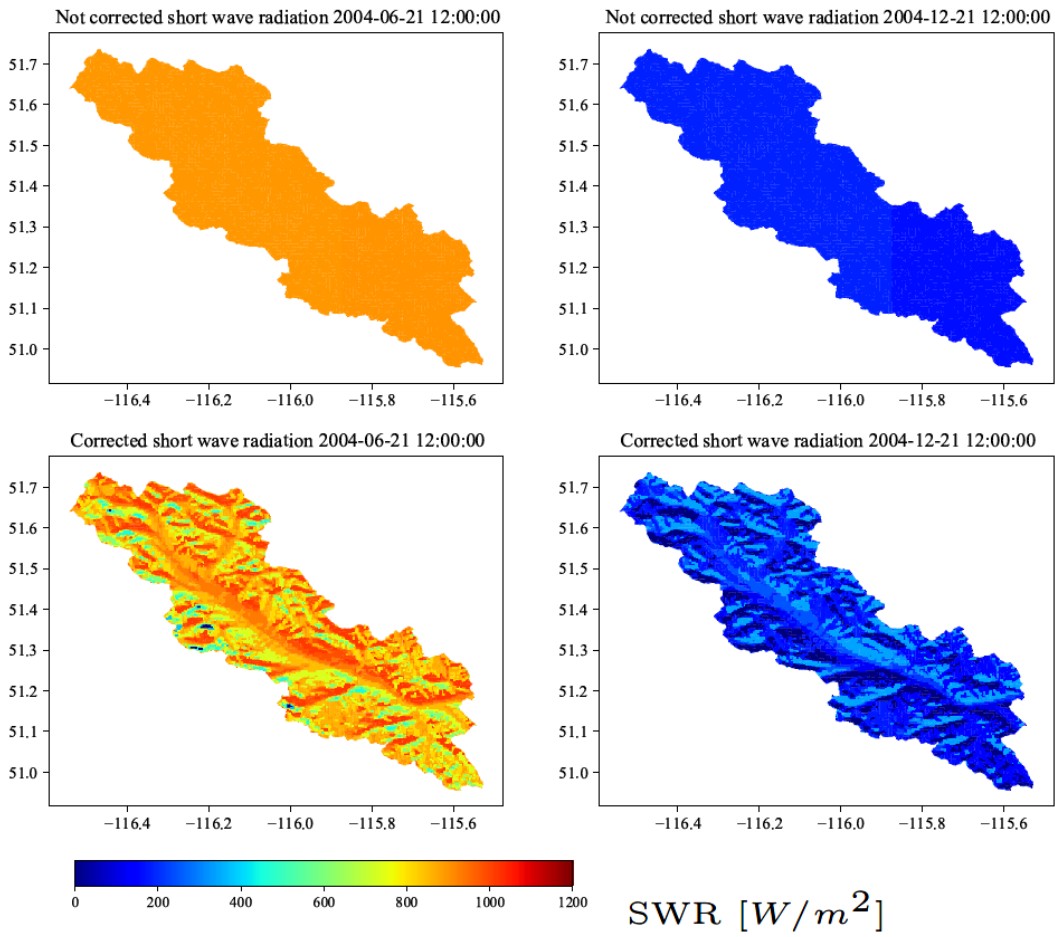


Figure A-1 Short wave radiation for (top left) not corrected for slope and aspect and (bottom left) corrected for slope and aspect for 21st June 2020 and (top right) not corrected for slope and ascpet and (bottom right) corrected for slope and aspect for 21st December 2020.
