# Peer review of "Flexible vector-based spatial configurations in land models"

_Hydrology and Earth System Sciences, 2020_

## Referee Comment (RC1) · Anonymous Referee #1 · 29 Apr 2020

Gharari et al present an application of the VIC model, using Grouped Response Units to define computational units, rather than grids. It is acknowledged that this concept was already presented in 1993. I do think it is justified to re-introduce older concepts if these can serve the science of today, however, then the re-introduction should also deal with some of the challenges of today, and this is currently not the case.

Firstly, the reader has to do quite some searching to fully capture the concept of GRU's, and its comparison to HRU's. Only when the investigated cases are presented it becomes clear what a GRU exactly is and the choices it encompasses when defining GRU's. This seems to be the result of an overall quite weak structure in the manuscript; the introduction does not clearly present the aim or goal, probably because the structural test (case 3, presented in the intro in line 103-110) seems to be completely out

of context. In the same fashion, 3.1.3 is not well embedded. Furthermore, sections are not logically structured, e.g. subsection 3.3 only consists of 2 sentences while some subsubsections are longer, and parameter are presented well after the calibration is discussed and the cases are introduced. I suggest restructuring the manuscript, clearly introducing the concepts with simple examples, and omitting parts that do not fit the aim or goal of the study.

One of the key questions in defining the spatial discretization of models is of course the calibration. Whereas the GRU's conceptually might make sense compared to grid cells, it introduces new questions on how to calibrate the parameters, and this is not well explained in the text. Does each GRU receive its own set of parameters? And is this then related in any way to the underlying data? As the authors rightly suggest, parameter ranges can be adapted based on soil type or land use, but it seems this was not done by the authors. Not surprisingly, the results demonstrate some of the already known flaws from calibrating on discharge outlet; the everlasting problem of equifinality and overparameterization. If the authors believe the GRU concept is valuable to re-introduce (and I can see it has potential), this value should be demonstrated in a more sophisticated calibration. If the same calibration is done as for usual grid-models, of course we know we can achieve good model performance because there are enough buttons to push, but what do we learn from it compared to a grid-based model and what does it add? 1000 evaluations in the calibration procedure seems rather limited given the dimensions of the problem; this is understandable from a computational point of view, but also a chance to demonstrate why GRU's make more sense than grids within these bounds, by making use of the opportunities that GRU's offer in comparison to grids.

An example: The results from Figure 4 are criticized in the text as: "The result indicates the two parameters that are often fixed or a priori allocated based on look up tables can exhibit significant uncertainty and non-identifiability". The Brooks-Corey coefficient is from such a high conceptual level that it might be challenging to find good values

in lookup tables, but Ksat might be able to be estimated. The lookup tables can then provide an indication for a search range for the parameter and decrease the equifinality issues with these two parameters.

Shortly, I can see why GRU's might have added value in land-surface modeling. However, the re-introduction of this concept in this manuscript might not make a very good case to convince people of this fact, given that calibration is one of the main challenges and the potential for GRU's in this context is not well explored.

Other suggestions:

In section 2.3, it remains unclear why structural changes to the model were made. Some of the most sensitive parameters of the model (Ds, Dm) have been replaced by a linear reservoir coefficient. Furthermore, the description focusses on VIC4 while VIC5 was explored. Why is that?

It is not explained how the parameters in Table 2 were selected for calibration. It is for instance remarkable that no snow parameters, such as snow roughness, are included in the calibration – is this because GRU's focus on soil and land use? Furthermore, it is not clarified to which soil layer Eexp and Ksat refer, or is this kept constant over both soil layers?

Minor for tables and figures:

Table 1 the unit of forcing resolution is missing (degree) Figure 3 the a,b,c labels are missing, the legend is not readable. Figure 4 not sure if this is very informative. More interesting to see a boxplot of every parameter to demonstrate the wide range. Figure 5 Caption says "deviation" but you demonstrate NSE compared to benchmark run, and not the deviation in NSE.

---

## Author Comment (AC1) · 4 May 2020

**Answer to the comments by anonymous reviewer#1**

We thank the reviewer for their constructive comments on our work. For a fruitful open discussion and to clarify some points we have drafted this early reply to the reviewer. A more detailed reply will follow the revised version identifying point by point changes to the manuscript based on reviewer's comments/suggestions. For convenience, the reviewer's comments are given in blue.

Gharari et al present an application of the VIC model, using Grouped Response Units to define computational units, rather than grids. It is acknowledged that this concept was already presented in 1993. I do think it is justified to re-introduce older concepts if these can serve the science of today, however, then the re-introduction should also deal with some of the challenges of today, and this is currently not the case.

As the reviewer rightly mentioned, this manuscript does not intend to introduce the concept of GRU, instead it tries to used it as a base for implementation of the VIC model which has been used worldwide and hopefully draw the attention for wider land model community to use the concept of vector-based setup based on GRUs.

Solving the "challenges of today" (which the reviewer describes in more depth in their later comments, and which we respond to in more depth later in this document) is not the main goal of this paper. In this manuscript, we try to point out the technical and scientific advantages of using a vector-based setup based on the concept of GRU. To do so, here we reflect on one of often not very well explored challenges in land modeling community, that is trade-off between accuracy of the land models' spatial representation and their performance. We hope our paper sheds some light on the ongoing discussion. The GRU concept was very helpful in this respect as we could change the resolution of forcing without really affecting the parameter values at the GRU level as there is no upscaling to the grid resolution. We think that is major advantage that we highlighted in this manuscript.

Firstly, the reader has to do quite some searching to fully capture the concept of GRU's, and its comparison to HRU's. Only when the investigated cases are presented it be- comes clear what a GRU exactly is and the choices it encompasses when defining GRU's. This seems to be the result of an overall quite weak structure in the manuscript; the introduction does not clearly present the aim or goal, probably because the structural test (case 3, presented in the intro in line 103-110) seems to be completely out of context. In the same fashion, 3.1.3 is not well embedded. Furthermore, sections are not logically structured, e.g. subsection 3.3 only consists of 2 sentences while some subsubsections are longer, and parameter are presented well after the calibration is discussed and the cases are introduced. I suggest restructuring the manuscript, clearly introducing the concepts with simple examples, and omitting parts that do not fit the aim or goal of the study.

We agree with the reviewer that the introduction structure can be improved. We agree that the third experiment is somewhat out of the scope of this paper (although experiment three is related to the parameter/process uncertainty; will be mentioned in the following). We will revise this

manuscript, clarify the concept of the GRU earlier in the manuscript and remove the experiment three (or perhaps move it to an appendix and refer to it shortly).

One of the key questions in defining the spatial discretization of models is of course the calibration. Whereas the GRU's conceptually might make sense compared to grid cells, it introduces new questions on how to calibrate the parameters, and this is not well explained in the text. Does each GRU receive its own set of parameters? And is this then related in any way to the underlying data? As the authors rightly suggest, parameter ranges can be adapted based on soil type or land use, but it seems this was not done by the authors. Not surprisingly, the results demonstrate some of the already known flaws from calibrating on discharge outlet; the everlasting problem of equifinality and overparameterization. If the authors believe the GRU concept is valuable to reintroduce (and I can see it has potential), this value should be demonstrated in a more sophisticated calibration. If the same calibration is done as for usual grid-models, of course we know we can achieve good model performance because there are enough buttons to push, but what do we learn from it compared to a grid-based model and what does it add? 1000 evaluations in the calibration procedure seems rather limited given the dimensions of the problem; this is understandable from a computational point of view, but also a chance to demonstrate why GRU's make more sense than grids within these bounds, by making use of the opportunities that GRU's offer in comparison to grids.

We thank the reviewer for this comment. We fully agree with the reviewer's comments on the parameters' values and estimation. Everything boils down to how GRUs are parameterized. We will better explain our approach to GRU parametrization during our revisions. We will explain how the parameters for each GRU are derived and how this relates to available data. Just a brief explanation here:

1- The soil layers get the same set of parameters for bulk density, saturated hydraulic conductivity (no difference between the vertical soil layers).
2- The conceptual soil parameters such as binf are unified across the scale (similar to most of VIC application).
3- The soil depth that conceptually define the storage of the system are defined based on land cover. The forested areas have deeper soil (or root zone) to allow for larger storage and transpiration. Just to mention that is this is an advantage of the vector-based implementation.
4- $K_{slow}$ is similar for the entire system (or a gauges) as it can be inferred/calibrated only from the recession analysis.

Of course, more intuitive and sophisticated parameter allocation can be explored but the above-mentioned parameter selection is aligned with what is often done for calibrating the VIC model. This is purposefully not to make the regionalisation so complex that the manuscript deviates from its own message (which is vector-based implementation and accuracy-trade-off implementation).

I know my colleagues who work with MESH model sometime do this distinction between parameter values of various GRUs in their applications/scientific explorations, for example, different soil with different land cover have different parameters. I personally do not move to

that direction for few reasons: (1) the parameters of the spatially largest GRU will be the most sensitive ones when calibrating against the observed streamflow (or polishing of smaller GRUs that have very small contribution, <1%, may be needed), (2) expansion of parameter for calibration that we don't know how to tied actually will unnecessarily add to the dimension of the problem (no information tangible construct them). There is ongoing effort to relate the parameters to physical characteristics but each of the decisions in itself is an assumption and cannot be inferred directly from the data (for example Mizukami et al., 2017 Table-3). We totally agree with the reviewer on "this value should be demonstrated in a more sophisticated calibration" but at the same time we have not much data for the sophisticated calibration especially the entire subsurface flow movement.

We should also emphasize that part of the motivation for GRU is computational efficiency with respect to optimization, for example in the MESH model. The underlying assumption is that grouping units from a parametrization perspective, since we expect them to behave in a physical similar way, allows us to characterize the variability with respect to the forcing and of course the subsequent hydrological response, while maintaining the degrees of freedom for parameter estimation reasonable. Our accuracy-performance trade-off is aligned with this mentality also.

Reflecting on reviewer comment, there might be two benefits of the GRU implementation:

1- Technical aspect; which is the ease of parameter allocation to a GRU (as each GRU has a specific land cover and soil type), or better implementation of regionalization rules if applicable. Easier coupling with vector-based routing.
2- The scientific values of implementing the models in a GRU approach. That is a grand challenge and an ongoing development. For example, how to effectively parameterize the model simulation at GRUs. I personally think part of the reason that the GRU was overlooked or not implemented widely for land models, as the reviewer mentioned with sophisticated calibration, is the lack of data and proper understanding of how parameters behave at the scale of modeling. One of the scientific applications we had here is the trad-off between accuracy of spatial representation and model performance.

Perhaps, I agree that the current manuscript has more emphasis on the first than the latter but at the same time the GRU implementation can be vehicle to explore the more scientific questions. For sake of simplicity and to emphasis on the advantages of the GRU implementation, we tried to put less emphasis on the sophisticated parameterization. As computational hydrology team at University of Saskatchewan we are moving to face this grand challenge in future.

The modeling set up, calibration and parameter perturbation for calibration are based on the objective of the modeling. Our objective here is efficiently-accuracy trade-off for land model and hydrograph simulation.

Reflecting on the computational costs of the setups:

1- 1000 simulations are selected as an arbitrary value we chose based on the computational infrastructure (and time) we had available for the Case4-4km which has 6000 computational units (the most computationally expensive setup).

2- For context, 6000 computational units will be equivalent of approximately 1/3 of the CONUS domain with standard gridded simulation at 0.25 by 0.25 degree lat/lon.
3- Consequently, running 1000 simulations for the Case4-4km takes 5 days on 50 CPUs of ComputeCanada infrastructure (assuming nothing goes wrong with the job), or approximately 8 months on a single CPU.
4- We have tried the impact of longer calibration runs for simpler cases (up to couple of thousands of simulations) but did not find a noticeable increase in NS scores.
5- Also, our choice for hourly forcing data increases the storage space needed by almost a factor 40 compared to daily inputs that is used for VIC-4 and earlier.

An example: The results from Figure 4 are criticized in the text as: "The result indicates the two parameters that are often fixed or a priori allocated based on look up tables can exhibit significant uncertainty and non-identifiability". The Brooks-Corey coefficient is from such a high conceptual level that it might be challenging to find good values in lookup tables, but Ksat might be able to be estimated. The lookup tables can then provide an indication for a search range for the parameter and decrease the equifinality issues with these two parameters.

The soil data that need to inform this choice are themselves so uncertain that they may not guarantee more "realistic $k_{sat}$ ranges". Also sub-resolution heterogeneity is not account for, nor is $k_{sat}$ very relevant if the dominant flowpath is macropores (which might be very well the case given the significant elevation differences in the region of study). Therefore, we didn't try to a priori limit the $k_{sat}$ calibration ranges too much. This uncertainty in $k_{sat}$ also has high implication for the future regionalization that might be built partly on $k_{sat}$.
We also wanted to indicate that parameters may be more uncertain that what is suggested in look up tables. It is often the case the land modeling community "kill" the potential uncertainty in parameters and processes either by assigning (hardcoded) parameter values from look up tables or using calibration techniques that yield single best solutions (Mendoza et al., 2014).

Shortly, I can see why GRU's might have added value in land-surface modeling. How- ever, the re-introduction of this concept in this manuscript might not make a very good case to convince people of this fact, given that calibration is one of the main challenges and the potential for GRU's in this context is not well explored.

We thank the reviewer for his/her constructive suggestions. We try to improve the manuscript flow. Meanwhile, we would like emphasis that the focus of this manuscript is not to come up with the parameterization scheme for the model but instead to provide an alternative representation of spatial data that can be from benefit for land modeling community. We demonstrated that how an existing model, such as VIC, can easily be implemented in a vector-based framework. Moreover, the vector-based implementation is not tied to any calibration strategy. A modeller may use the vector-based set up for a land model while avoid any automatic calibration.

Other suggestions:

In section 2.3, it remains unclear why structural changes to the model were made. Some of the most sensitive parameters of the model (Ds, Dm) have been replaced by a linear reservoir

coefficient. Furthermore, the description focusses on VIC4 while VIC5 was explored. Why is that?

It is a very good point for discussion.

1- VIC has a baseflow formulation that has 6 parameters, 4 for the baseflow formulation and 2 for the physical specification of the depth and porosity of the lower layer. These 6 parameters are impossible to be inferred from the recession analysis of a hydrograph.
2- The common structure of the land models does not allow for recession analysis based on master recession curve. The soil layers act like a cascade of reservoirs. The water movement from the second layer of VIC can already be damped enough that it may not need even need a slow baseflow reservoir (Gharari et al., 2019).
3- One simple solution to that is to basically seize the micropore water movement to the baseflow layer, and only allow macropore water movement based on fraction of surface runoff for example (experiment three).
4- The recession coefficient of the Bow River at Banff is in scale of 0.01 day$^{-1}$ or 100 days. This is similar to what we get from the automatic calibration. $K_{slow}$ is one identifiable and sensitive parameter given the regime of the Bow River.

Going back to the "sophisticated calibration". Here we are somehow calibrating the model structure based on the general hydrological knowledge that water movement is mostly preferential (macropore) in areas with high elevation differences. It can be assumed that the preferential flow pathways are activity passing the water in comparison with micropore water movement in the region of study that has such a huge elevation difference. Subsequently any inference or assumption on $K_{sat}$ based on an ill posed model structure may not be a valid assumption. This if one of the points the manuscript tries to make.

We mentioned VIC-4 to emphasis why VIC was used so widely. We will remove the explanation on VIC-4.

It is not explained how the parameters in Table 2 were selected for calibration. It is for instance remarkable that no snow parameters, such as snow roughness, are included in the calibration – is this because GRU's focus on soil and land use? Furthermore, it is not clarified to which soil layer Eexp and Ksat refer, or is this kept constant over both soil layers?

We have indeed chosen to only calibrate those parameters that relate to aspects of our GRU configuration (and more importantly forcing resolution for accuracy-efficiently trade-off). We will clarify this in the text. We also set the $K_{sat}$ and $E_{exp}$ similar for both layers. We will clarify that as well.

Minor for tables and figures:

Table 1 the unit of forcing resolution is missing (degree) Figure 3 the a,b,c labels are missing, the legend is not readable. Figure 4 not sure if this is very informative. More interesting to see a boxplot of every parameter to demonstrate the wide range. Figure 5 Caption says "deviation" but you demonstrate NSE compared to benchmark run, and not the deviation in NSE.

We will fix the forcing resolution in Table-1.

We will fix Figure-3.

We will replace figure 4 with a more informative possibly a box plot.

Please not that this is deviation from synthetic case and not standard deviation. We will rephrase this part.

We thank the reviewer for the constructive comments, and we hope to enrich our manuscript by addressing the reviewer's comment sufficiently and successfully.

With kind regards,

Shervan Gharari, on behalf of the co-authors

References:

Gharari, S., Clark, M., Mizukami, N., Wong, J.S., Pietroniro, A. and Wheater, H., 2019. Improving the representation of subsurface water movement in land models. *Journal of Hydrometeorology*, (2019).

Mendoza, P.A., Clark, M.P., Barlage, M., Rajagopalan, B., Samaniego, L., Abramowitz, G. and Gupta, H., 2015. Are we unnecessarily constraining the agility of complex process-based models?. *Water Resources Research*, *51*(1), pp.716-728.

Mizukami, N., Clark, M.P., Newman, A.J., Wood, A.W., Gutmann, E.D., Nijssen, B., Rakovec, O. and Samaniego, L., 2017. Towards seamless large-domain parameter estimation for hydrologic models. *Water Resources Research*, *53*(9), pp.8020-8040.

---

## Referee Comment (RC2) · Anonymous Referee #2 · 9 May 2020

The paper titled "Flexible vector-based spatial configurations in land models" uses a new spatial configuration approach with the VIC model that is based on the group response unit concept. The main goals in the paper are to first introduce a method to defining heterogeneity in VIC and then to assess the added value of multiple spatial configurations over the Bow River basin at Banff. The paper is a novel contribution and will be an excellent addition to the land surface/hydrologic modeling community. However, there are multiple issues that I describe below that should be addressed before publication.

* The abstract is too long. I strongly suggest reducing the size of the abstract by 20%-40%.

* I don't really understand the difference between GRU and HRU in this study; from my

understanding a GRU is composed of many HRUs. For example, a sub-basin (GRU) will have multiple HRUs. But based on what is being done here, these GRUs are just the classic GIS partitioning happening and thus very similar to the original definition of a HRU. Maybe I am misunderstanding something. In any case, please clarify the use of the GRU term here.

* Figures 1 and 2 - These two figures are not very informative—especially Figure 1. I would remove them. Maybe some of these ideas could be merged into an improved (or split) Figure 3.

* Figure 3 - You should have a, b, c, d coded on the panels themselves. Panel c is very unclear. I think this whole figure is critical to understand the implementation and thus should be improved (and perhaps split into two).

* Figure 6 - The 3 dimensionality of this figure is unnecessary and frankly confusing. The 2d surface is more than enough to get the point across.

* Figure 8 - Again, the 2d surface would be much better here.

* Line 122 - Only using tmin, tmax, precipitation, and wind speed is only one option in the earlier VIC versions. One could also still use longwave in, shortwave in, among others.

* Line 153 - Although I am certainly a fan of "killing the grid", it is not entirely true that "resolution loses its meaning" with the introduced approach. You still have an effective spatial resolution which is governed by the level of details that needs to exist in your polygons. Of course the advantage here is that you can have the size of those polygons vary as a function of space; however, you will still have the concept of an effective spatial resolution present. I'd suggest thinking more carefully of what moving to a polygon based approach really means and how it can be "upscaled" in more informative ways than the classic coarsening of the grid.

* Check for typos; there appear to be a few throughout the text (e.g., VIV-GRU on line

182)

---

## Author Comment (AC2) · 12 May 2020

**Answer to the comments by anonymous reviewer#2**

We thank the reviewer for their comments on our manuscript. This is a short reply to the two major comments raised by the reviewer. We tried to address them in this brief reply to stimulate open discussion. We agree with the reviewer on their other comments and we will provide detailed answers to their individual comments after the open discussion.

The paper titled "Flexible vector-based spatial configurations in land models" uses a new spatial configuration approach with the VIC model that is based on the group response unit concept. The main goals in the paper are to first introduce a method to defining heterogeneity in VIC and then to assess the added value of multiple spatial configurations over the Bow River basin at Banff. The paper is a novel contribution and will be an excellent addition to the land surface/hydrologic modeling community. However, there are multiple issues that I describe below that should be addressed before publication.

* I don't really understand the difference between GRU and HRU in this study; from my understanding a GRU is composed of many HRUs. For example, a sub-basin (GRU) will have multiple HRUs. But based on what is being done here, these GRUs are just the classic GIS partitioning happening and thus very similar to the original definition of a HRU. Maybe I am misunderstanding something. In any case, please clarify the use of the GRU term here.

We thank the reviewer for their comment. The reviewer is correct on the original concept of GRUs and HRUs. As the reviewer correctly points out, GRUs and HRUs typically define a hierarchal spatial organization where HRUs are nested within GRUs (e.g., Clark et al., 2015). For example, in land models, a GRU could be a model grid box and HRUs could be the vegetation tiles within a model grid box; in hydrology models, a GRU could be a sub-basin and HRUs the hydrologically similar areas within a sub-basin. The forcing data could be either constant across the HRUs or distributed to each HRU (distributed forcing within GRUs is important in cases where there are strong climate gradients within GRUs, e.g., due to large elevation range). GRUs are also used to describe classifications of the landscape across the modelling domain (Kouwen et al., 1993, Pietroniro et al., 2007).

We agree with the reviewer that using the vector-based implementation there is little difference between the concept of GRU and HRU. To avoid confusion, we abandon the concept of GRU entirely and present the model as a vector-based implementation (that can benefit from the concepts of GRU and HRU).

* Line 153 - Although I am certainly a fan of "killing the grid", it is not entirely true that "resolution loses its meaning" with the introduced approach. You still have an effective spatial resolution which is governed by the level of details that needs to exist in your polygons. Of course the advantage here is that you can have the size of those polygons vary as a function of space; however, you will still have the concept of an effective spatial resolution present. I'd suggest thinking more carefully of what moving to a polygon based approach really means and how it can be "upscaled" in more informative ways than the classic coarsening of the grid.

An excellent point raised by the reviewer. The reviewer is certainly correct that we still have the same upscaling challenges in vector-based implementations. We will discuss this issue in more detail in the revised paper.

One of the ideas behind the vector-based modeling is the flexibility of the input data. For example, for a larger basin, the forcing can be set to a higher resolution for the mountainous headwater. For our test case (limited spatial domain), this concept can be explored in detail, and our current work focuses on resampling and coarsening of the forcing grids. We will rework this section to include more discussion on the importance of "polygons" vs "grid" and the implications for large/continental scale modeling.

With kind regards,

Shervan Gharari, on behalf of the co-authors

**References:**

Clark, M.P., Nijssen, B., Lundquist, J.D., Kavetski, D., Rupp, D.E., Woods, R.A., Freer, J.E., Gutmann, E.D., Wood, A.W., Brekke, L.D. and Arnold, J.R.: A unified approach for process-based hydrologic modeling: 1. Modeling concept. *Water Resources Research*, *51*(4), pp.2498-2514, 2015.

Pietroniro, A., Fortin, V., Kouwen, N., Neal, C., Turcotte, R., Davison, B., Verseghy, D., Soulis, E. D., Caldwell, R., Evora, N., and Pellerin, P.: Development of the MESH modelling system for hydrological ensemble forecasting of the Laurentian Great Lakes at the regional scale, Hydrol. Earth Syst. Sci., 11, 1279–1294, https://doi.org/10.5194/hess-11-1279-2007, 2007.

Kouwen, N., Soulis, E.D., Pietroniro, A., Donald, J. and Harrington, R.A.: Grouped response units for distributed hydrologic modeling. *Journal of Water Resources Planning and Management*, *119*(3), pp.289-305, 1993.

---

## Author Response (AR1)

**Letter to the Editor:**

Dear Dr. Niko Wanders,

Thank you very much for handling our manuscript in HESSD and facilitating the fruitful open discussion process. We did our best to address reviewers' comments in the revised manuscript. The major changes are as follow:

1- We have reshuffled the structure of the manuscript so that the concept of vector-based configuration of land models is presented earlier and separately in the manuscript (currently in Section 2).
2- We have reworded the VIC-GRU to vector-based configuration of land models instead. We used the concept of the GRU for parameter allocation in this work [which can be different for other studies and parameter allocation].
3- We have re-calibrated the parameters with snow roughness as requested by the first reviewer.
4- We have merged two of the Figures and simplified the first Figure. The numbering of the Figures may have changed due to the change in the structure of the manuscript.
5- We fully removed Experiment-3 which was exploring the presence of the macropore water movement in the VIC model.

We hope the current changes in the manuscript satisfy the editor and the reviewers and hopefully warrant publication in the current format. Looking forward to hearing back from you.

With kind regards,

Shervan Gharari, on behalf of the co-authors

**Answer to the comments by anonymous reviewer#1**

We thank the reviewer for their constructive comments on our work. For convenience, the reviewer's comments are given in green and our response is in blue.

Gharari et al present an application of the VIC model, using Grouped Response Units to define computational units, rather than grids. It is acknowledged that this concept was already presented in 1993. I do think it is justified to re-introduce older concepts if these can serve the science of today, however, then the re-introduction should also deal with some of the challenges of today, and this is currently not the case.

As the reviewer rightly mentioned, this manuscript does not intend to introduce the concept of GRU, instead, it tries to use it as a base for implementation of the VIC model in a vector-based fashion which has been used worldwide and hopefully draw the attention for wider land model community to use the concept of vector-based setup (and based on GRUs parameterization).

Solving the "challenges of today" (which the reviewer describes in more depth in their later comments, and which we respond to in more depth later in this document) is not the main goal of this paper. In this manuscript, we try to point out the technical and scientific advantages of using a vector-based setup. To do so, here we reflect on one of often not very well explored challenges in land modeling community, that is trade-off between accuracy of the land models' spatial representation and their performance. We hope our paper sheds some light on the ongoing discussion. The vector-based implementation concept was very helpful in this respect as we could change the resolution of forcing without really affecting the parameter values at the computational units level as there is no upscaling to the grid resolution. It was also very helpful to use the GRU concept in the parameterization of the VIC model however other parameterization techniques could be explored within the vector-based implementation of land models. We think that is a major advantage that we highlighted in this manuscript. Also, following the suggestion by the second reviewer, we have moved the use of GRU concept in grouping the parameter values for a specific combination of geospatial data after introducing the vector-based implementation concept.

Firstly, the reader has to do quite some searching to fully capture the concept of GRU's, and its comparison to HRU's. Only when the investigated cases are presented it be- comes clear what a GRU exactly is and the choices it encompasses when defining GRU's. This seems to be the result of an overall quite weak structure in the manuscript; the introduction does not clearly present the aim or goal, probably because the structural test (case 3, presented in the intro in line 103-110) seems to be completely out of context. In the same fashion, 3.1.3 is not well embedded. Furthermore, sections are not logically structured, e.g. subsection 3.3 only consists of 2 sentences while some subsubsections are longer, and parameter are presented well after the calibration is discussed and the cases are introduced. I suggest restructuring the manuscript, clearly introducing the concepts with simple examples, and omitting parts that do not fit the aim or goal of the study.

We agree with the reviewer that the introduction structure can be improved (this is also mentioned by the second reviewer). We agree that the third experiment is somewhat out of the scope of this paper (although experiment three is related to the parameter/process uncertainty; will be mentioned in the following). We removed Experiment-3. We have reworked the structure so that the vector-based implementation of land models comes earlier as a separated section (Section 2). We hope the current structure is easier to read and follow.

One of the key questions in defining the spatial discretization of models is of course the calibration. Whereas the GRU's conceptually might make sense compared to grid cells, it introduces new questions on how to calibrate the parameters, and this is not well explained in the text. Does each GRU receive its own set of parameters? And is this then related in any way to the underlying data? As the authors rightly suggest, parameter ranges can be adapted based on soil type or land use, but it seems this was not done by the authors. Not surprisingly, the results demonstrate some of the already known flaws from calibrating on discharge outlet; the everlasting problem of equifinality and overparameterization. If the authors believe the GRU concept is valuable to reintroduce (and I can see it has potential), this value should be demonstrated in a more sophisticated calibration. If the same calibration is done as for usual grid-models, of course we know we can achieve good model performance because there are enough buttons to push, but what do we learn from it compared to a grid-based model and what does it add? 1000 evaluations in the calibration procedure seems rather limited given the dimensions of the problem; this is understandable from a computational point of view, but also a chance to demonstrate why GRU's make more sense than grids within these bounds, by making use of the opportunities that GRU's offer in comparison to grids.

We thank the reviewer for this comment. We fully agree with the reviewer's comments on the parameters' values and estimation. Everything boils down to how computational units are parameterized (which can be very well based on the GRU concept or other techniques). We tried to briefly explain this in the VIC parameter specification in Section 3.3.1. Just a brief explanation here:

1- The soil layers get the same set of parameters for bulk density, saturated hydraulic conductivity (no difference between the vertical soil layers). This is added to Section 3.3.1.
2- The conceptual soil parameters such as $b_{inf}$ are unified across the scale (similar to most of VIC application).
3- The soil depths that conceptually define the storage of the system are defined based on land cover. The forested areas have deeper soil (or root zone) to allow for larger storage and transpiration. Just to mention that this is an advantage of the vector-based implementation and an illustration of more sophisticated calibration strategy. If more hydrological knowledge at the scale of interest is available that can be translated into the constraint.
4- $K_{slow}$ is similar for the entire system (or a gauges) as it can be inferred/calibrated only from the recession analysis.

Of course, more intuitive and sophisticated parameter allocation can be explored but the above-mentioned parameter selection is aligned with what is often done for calibrating the VIC model. This is purposefully not to make the regionalisation so complex that the manuscript deviates from its own message (which is vector-based implementation and accuracy-efficiency trade-off implementation).

I know my colleagues who work with MESH model sometime do this distinction between parameter values of various GRUs in their applications/scientific explorations, for example, different soil with different land cover have different parameters. I personally do not move to that direction for few reasons: (1) the parameters of the spatially largest GRU will be the most sensitive ones when calibrating against the observed streamflow (or polishing of smaller GRUs that have very small contribution, <1%, may be needed), (2) expansion of parameter for calibration that we don't know how to tied actually will unnecessarily add to the dimension of the problem (no information tangible construct them). There is an ongoing effort to relate the parameters to physical characteristics but each of the decisions in itself is an assumption and cannot be inferred directly from the data (for example Mizukami et al., 2017 Table-3). We totally agree with the reviewer on "this value should be demonstrated in a more sophisticated calibration" but at the same time we have not much data for the sophisticated calibration especially the entire subsurface flow movement.

We should also emphasize that part of the motivation for vector-based configuration is computational efficiency with respect to optimization, for example in the MESH model, the underlying assumption is that grouping units from a parametrization perspective, since we expect them to behave in a physical similar way, allows us to characterize the variability with respect to the forcing and of course the subsequent hydrological response, while maintaining the degrees of freedom for parameter estimation reasonable. Our accuracy-performance trade-off is aligned with this mentality also.

Reflecting on reviewer comment, there might be two benefits of allocating the computational units parameter values:

1- Technical aspect; which is the ease of parameter allocation to a computational unit (as each computational unit has a specific land cover and soil type), or better implementation of regionalization rules if applicable. Easier coupling with vector-based routing.
2- The scientific values of implementing the models in a vector-based fashion. That is a grand challenge and an ongoing development. For example, how to effectively parameterize the model simulation at computational units. I personally think part of the reason that the GRU parameter allocation was overlooked or not implemented widely for land models, as the reviewer mentioned with sophisticated calibration, is the lack of data and proper understanding of how parameters behave at the scale of modeling. One of the scientific applications we had here is the trade-off between accuracy of spatial representation and model performance.

Perhaps, I agree that the current manuscript has more emphasis on the first than the latter but at the same time the vector-based implementation can be a vehicle to explore the more scientific questions. For sake of simplicity and to emphasize on the advantages of the vector-based configuration, we tried to put less emphasis on the sophisticated parameterization. As computational hydrology team at University of Saskatchewan we are moving to face this grand challenge in future.

The modeling set up, calibration and parameter perturbation for calibration are based on the objective of the modeling. Our objective here is accuracy-efficiency trade-off for land model and hydrograph simulation.

Reflecting on the computational costs of the setups:

1- 1000 simulations are selected as an arbitrary value based on the computational infrastructure (and time) we had available for the Case4-4km which has 6000 computational units (the most computationally expensive setup).
2- For context, 6000 computational units will be equivalent of approximately 1/3 of the CONUS domain with standard gridded simulation at 0.25 by 0.25 degree lat/lon.
3- Consequently, running 1000 simulations for the Case4-4km takes 5 days on 50 CPUs of ComputeCanada infrastructure (assuming nothing goes wrong with the job), or approximately 8 months on a single CPU.
4- We have tried the impact of more calibration runs for simpler cases (up to couple of thousands of simulations) but did not find a noticeable increase in NS scores.
5- Also, our choice for hourly forcing data increases the storage space needed by almost a factor of 40 compared to daily inputs that is used for VIC-4 and earlier.

An example: The results from Figure 4 are criticized in the text as: "The result indicates the two parameters that are often fixed or a priori allocated based on look up tables can exhibit significant uncertainty and non-identifiability". The Brooks-Corey coefficient is from such a high conceptual level that it might be challenging to find good values in lookup tables, but Ksat might be able to be estimated. The lookup tables can then provide an indication for a search range for the parameter and decrease the equifinality issues with these two parameters.

The soil data that need to inform this choice are themselves so uncertain that they may not guarantee more "realistic $k_{sat}$ ranges". Also, sub-resolution heterogeneity is not account for, nor is $k_{sat}$ very relevant if the dominant flowpath is macropores (which might be very well the case given the significant elevation differences in the region of study). Therefore, we didn't try to a priori limit the $k_{sat}$ calibration ranges too much. This uncertainty in $k_{sat}$ also has high implication for the future regionalization that might be built partly on $k_{sat}$.
We also wanted to indicate that parameters may be more uncertain that what is suggested in look up tables. It is often the case the land modeling community "kill" the potential uncertainty in parameters and processes either by assigning (hardcoded) parameter values from look up tables or using calibration techniques that yield single best solution (Mendoza et al., 2014).

Shortly, I can see why GRU's might have added value in land-surface modeling. How- ever, the re-introduction of this concept in this manuscript might not make a very good case to convince people of this fact, given that calibration is one of the main challenges and the potential for GRU's in this context is not well explored.

We thank the reviewer for his/her constructive suggestions. We try to improve the manuscript flow. Meanwhile, we would like to emphasize that the focus of this manuscript is not to come up with the parameterization scheme for the model but instead to provide an alternative representation of spatial data that can be beneficial for land modeling community. We demonstrated that how an existing model, such as VIC, can easily be implemented in a vector-based framework. Moreover, the vector-based implementation is not tied to any calibration strategy. A modeller may use the vector-based set up for a land model while avoiding any automatic calibration.

Other suggestions:

In section 2.3, it remains unclear why structural changes to the model were made. Some of the most sensitive parameters of the model (Ds, Dm) have been replaced by a linear reservoir coefficient. Furthermore, the description focusses on VIC4 while VIC5 was explored. Why is that?

It is a very good point for discussion.

1- VIC has a baseflow formulation that has 5 (or one can say 6) parameters, 4 for the baseflow formulation and 2 for the physical specification of the depth and porosity of the lower layer. These 6 parameters are impossible to be inferred from the recession analysis of a hydrograph.
2- The common structure of the land models does not allow for recession analysis based on master recession curve. The soil layers act like a cascade of reservoirs. The water movement from the second layer of VIC can already be damped enough that it may not even need a slow baseflow reservoir (Gharari et al., 2019).
3- One simple solution to that is to basically seize the micropore water movement to the baseflow layer, and only allow macropore water movement based on fraction of surface runoff for example (experiment three).
4- The recession coefficient of the Bow River at Banff is in scale of 0.01 day$_{-1}$ or 100 days. This is not similar to what we get from the automatic calibration. $K_{slow}$ is one identifiable and sensitive parameter given the regime of the Bow River but much higher value [this was wrongly stated in our first reply]. Similarly, the $D_s$ and $D_{smax}$ can show sensitivity but are they really identifiable if a simpler one-parameter baseflow does not hydrologically and logically does what is it expected to do [to simulate the baseflow and get close to the recession analysis]. These are the diagnostics checks that should be done even before any sensitivity analysis.

We mentioned VIC-4 to emphasize why VIC was used so widely. We will remove the explanation on VIC-4. We simplified the VIC description and focused only on VIC version 5.

It is not explained how the parameters in Table 2 were selected for calibration. It is for instance remarkable that no snow parameters, such as snow roughness, are included in the calibration – is this because GRU's focus on soil and land use? Furthermore, it is not clarified to which soil layer Eexp and Ksat refer, or is this kept constant over both soil layers?

We have indeed chosen to only calibrate those parameters that relate to aspects of computational units' configuration (and more importantly forcing resolution for accuracy-efficiency trade-off). We also set the $K_{sat}$ and $E_{exp}$ similar for both layers (clarified in the Section 3.3.1).

Additionally, and based on the reviewer's suggestion we included the snow roughness parameter in our calibration.

Minor for tables and figures:

Table 1 the unit of forcing resolution is missing (degree) Figure 3 the a,b,c labels are missing, the legend is not readable. Figure 4 not sure if this is very informative. More interesting to see a boxplot of every parameter to demonstrate the wide range. Figure 5 Caption says "deviation" but you demonstrate NSE compared to benchmark run, and not the deviation in NSE.

We fixed the forcing resolution in original Table-1 (which is now Table-2).

We fixed original Figure-3 (which is now Figure-2).

We replaced Figure-4 with a normalized illustration of the parameter values above $E_{NS}$ of 0.7. We added extra interpretation of this Figure in the Results and Discussion Sections.

Please not that this is deviation from synthetic case and not standard deviation. We rephrased the caption.

We thank the reviewer for the constructive comments, and we hope to enrich our manuscript by addressing the reviewer's comment sufficiently and successfully.

With kind regards,

Shervan Gharari, on behalf of the co-authors

**Answer to the comments by anonymous reviewer#2**

We thank the reviewer for their comments on our manuscript. For convenience, the reviewer's comments are given in black and our response is in blue.

The paper titled "Flexible vector-based spatial configurations in land models" uses a new spatial configuration approach with the VIC model that is based on the group response unit concept. The main goals in the paper are to first introduce a method to defining heterogeneity in VIC and then to assess the added value of multiple spatial configurations over the Bow River basin at Banff. The paper is a novel contribution and will be an excellent addition to the land surface/hydrologic modeling community. However, there are multiple issues that I describe below that should be addressed before publication.

The abstract is too long. I strongly suggest reducing the size of the abstract by 20%- 40%.

We have reduced the length of the abstract to ~75% of the original abstracts.

I don't really understand the difference between GRU and HRU in this study; from my understanding a GRU is composed of many HRUs. For example, a sub-basin (GRU) will have multiple HRUs. But based on what is being done here, these GRUs are just the classic GIS partitioning happening and thus very similar to the original definition of a HRU. Maybe I am misunderstanding something. In any case, please clarify the use of the GRU term here.

We thank the reviewer for their comment. The reviewer is correct on the original concept of GRUs and HRUs. As the reviewer correctly points out, GRUs and HRUs typically define a hierarchal spatial organization where HRUs are nested within GRUs (e.g., Clark et al., 2015). For example, in land models, a GRU could be a model grid box and HRUs could be the vegetation tiles within a model grid box; in hydrology models, a GRU could be a sub-basin and HRUs the hydrologically similar areas within a sub-basin. The forcing data could be either constant across the HRUs or distributed to each HRU (distributed forcing within GRUs is important in cases where there are strong climate gradients within GRUs, e.g., due to large elevation range). GRUs are also used to describe classifications of the landscape across the modelling domain (Kouwen et al., 1993, Pietroniro et al., 2007).

We agree with the reviewer that using the vector-based implementation there is little difference between the concept of GRU and HRU. To avoid confusion, we abandon the concept of GRU entirely and present the model as a vector-based implementation (that can benefit from the concepts of GRU and HRU). We only use the concept of GRU as a matter of parameter allocation in Section 3.3.1 and in the discussion to reflect on the validity of the assumption on parameter allocation.

Figures 1 and 2 - These two figures are not very informativeâATespecially Figure 1. I would remove them. Maybe some of these ideas could be merged into an improved (or split) Figure 3.

We have revised Figure 1 and made it more conceptual in the newer version. We have removed figure 2 to the discussion as an example the unrealistic combination of land cover and elevation zone in the VIC model for discussion purposes.

Figure 3 - You should have a, b, c, d coded on the panels themselves. Panel c is very unclear. I think this whole figure is critical to understand the implementation and thus should be improved (and perhaps split into two).

A, b, c, d are added to the panels. We have clarified the c as well to have a more readable legend. We believe the changes we made in Figure 1 now make the concept easier to understand. Therefore Figure 3 (now figure 2) is more directed to illustrates the basin of interest characteristics.

Figure 6 - The 3 dimensionality of this figure is unnecessary and frankly confusing. The 2d surface is more than enough to get the point across.

We have made it 2 D figure and merge it with original Figure 8.

Figure 8 - Again, the 2d surface would be much better here.

We have made it 2 D figure and merge it with original Figure 6.

Line 122 - Only using tmin, tmax, precipitation, and wind speed is only one option in the earlier VIC versions. One could also still use longwave in, shortwave in, among others.

We have revised the VIC description and now the description should be much clearer.

Line 153 - Although I am certainly a fan of "killing the grid", it is not entirely true that "resolution loses its meaning" with the introduced approach. You still have an effective spatial resolution which is governed by the level of details that needs to exist in your polygons. Of course the advantage here is that you can have the size of those polygons vary as a function of space; however, you will still have the concept of an effective spatial resolution present. I'd suggest thinking more carefully of what moving to a polygon based approach really means and how it can be "upscaled" in more informative ways than the classic coarsening of the grid.

An excellent point raised by the reviewer. The reviewer is certainly correct that we still have the same upscaling challenges in vector-based implementations. We discussed this issue in more detail in the revised paper (Section 2).

One of the ideas behind the vector-based modeling is the flexibility of the input data. For example, for a larger basin, the forcing can be set to a higher resolution for the mountainous headwater. For our test case (limited spatial domain), this concept can be explored in detail, and our current work focuses on resampling and coarsening of the forcing grids. We will rework this section to include more discussion on the importance of "polygons" vs "grid" and the implications for large/continental scale modeling.

* Check for typos; there appear to be a few throughout the text (e.g., VIV-GRU on line 182)

We tried to remove the typos as much as possible in the manuscript.

We thank the reviewer for the constructive comments, and we hope to enrich our manuscript by addressing the reviewer's comment sufficiently and successfully.

With kind regards,

Shervan Gharari, on behalf of the co-authors

**References:**

[revised manuscript text omitted]

**Moved up [17]:** <#>The diffusive wave parameters are set

**Moved up [16]:** <#>Impulse Response Function (IRF)

**Moved (insertion) [15]**

[revised manuscript text omitted]

---

## Author Response (AR2)

**Letter to the Editor:**

Dear Dr. Wanders

Once again thank you for handling our manuscript in Hydrology and Earth System Sciences.

We tried our best to answer the comments of the anonymous reviewer to the best we could, and we made necessary changed in the manuscript.

We look forward to your editorial decision on this manuscript.

With kind regards, Shervan Gharari, on behalf of co-authors

**Answer to the comments:**

Gharari et al did a good job in incorporating the feedback from the reviewers. The clarity and organisation of the paper has improved substantially. Especially Section 2, which now clearly explains the concept, and Figure 1 which demonstrates the concept, is a large improvement. Two main questions remain after reading the manuscript;

We thank the reviewer for their constructive comments on out work. Our point by point response is presented in blue in the following.

1) How does a vector-based spatial configuration compare to the 'classical' grid-based approach? This is not quantified, so probably the authors want to focus only on this difference at the conceptual level.

We thank the reviewer for their comments. As the reviewer mentioned we did not compare our model with grid-based VIC. We have tried to keep the comparison in the conceptual level. For example, the last figure, Figure-7, is comparing the grid vs vector-based implementation of VIC in conceptual case. please direct us to those lines if this comparison may have stated strongly and beyond the conceptual level.

2) The authors disagreed with my comment in the previous round that todays challenges related to calibration are not widely addressed, stating that it was not the aim of the paper to address current day challenges. Well, that is up to the authors to decide and not a content-based discussion where I as a reviewer should have a strong opinion about, but it leaves a bit of an unsatisfying feeling with the reader to find out only in line 502 that actually GRUs were calibrated, and that the challenges in defining GRUs remain with this vector-based approach.

We have tried to make it clear in the methodology section that the current vector-based approach does not solve the issue of model parameter allocation and identification (refer to your second minor point also)

Minor suggestions;

Introduction does not really converge towards vector-based spatial configuration. Again, not content based so not my role as a reviewer, but my impression as a reader.

The reviewer has a point here but the issue with lack of existing literature on systematic implementation of vector-based application makes creating that direction a bit challenging. We would be happy if the reviewer can direct us to some studies that indeed can funnel the introduction better to the vector-based implementation of land models.

Mention already in the methods that GRUs are calibrated and not every individual computational unit.

We have clarified point 4 in the methodology subsection. This point is about the ease of parameter allocation to each unit and not indicating that computational unit should be given parameters based on GRUs. We have tried to make this clearer in point 4 by adding a sentence.

Figure 1; perhaps indicate with numbers the 28 computational units? Or maybe a few, to further clarify the concept? Although it was clear for me already.

We thank the reviewer for this comment. We have added the numbers and the figure looks more interesting! Thank you.

Units of snow roughness in Table 1; shouldn't it be cm?

To my personal knowledge snow roughness is in order of few millimetres.

in general; it is not clarified or defended why these parameters were selected for calibration. It aren't necessarily the parameters that are identified as most sensitive in many other studies.

We thank the reviewer for their comment. We have investigated a sensitivity analysis on this case study with much more parameters than what is calibrated here. Similar to this study we found that the parameters are non-identifiable and there is significant interaction between them. One of the parameters that is among sensitive models and is not calibrated here are the routing parameter. Here we keep the routing parameter the same for all the model configurations so that comparison of the parameters across configurations are only the result of changes in forcing aggregations or parameter value at computational units excluding the change in routing parameters across configuration.

Figure 3a: Why not just show a boxplot per parameter? There seems to be no reason to connect the lines of the different parameters.

We have changed Figure 3a to boxplot. Thank you.

Figure 4: It would greatly help the reader if a summary of the four cases is added to the figure headers. Now, the reader has to search back in the text.

Very good suggestion we have added that to Figures-4 and 5. Now they can be easier understood even without caption.

Same for Figure 6; to help the reader, it could be indicated in the different blocks which information was adapted (snow for instance shows a clear effect of elevation, this would further stress/clarify that).

We have added titles to the Figure-6.

There are now double dots after every section header, I think this is not in line with the journal format.

We thank the reviewer. We have set this during the typesetting.

Language/grammar check, check for instance lines 33, 57, 150, 221, 343, 351, 492 (inconsistent symbols).

We have tried to refine the language in those lines that the reviewer mentioned (although we were not fully sure for few of them such as line 492 for example). We will also use the copyediting service of Copernicus Publication during typesetting.

Once again, we thank the reviewer for their constructive comments.

With kind regards, Shervan Gharari, on behalf of co-authors

| 1                                                  | Flexible vector-based spatial configurations in land models                                                                                                                                                                                                                                                                                                                                                                                                                                                                                                                                                                                                                                                                                                                                                                                                                                                  | Formatted: Font color: Text 1 |
|----------------------------------------------------|--------------------------------------------------------------------------------------------------------------------------------------------------------------------------------------------------------------------------------------------------------------------------------------------------------------------------------------------------------------------------------------------------------------------------------------------------------------------------------------------------------------------------------------------------------------------------------------------------------------------------------------------------------------------------------------------------------------------------------------------------------------------------------------------------------------------------------------------------------------------------------------------------------------|-------------------------------|
| 23                                                 |  <li>Shervan Gharari1,*, Martyn P. Clark1, Naoki Mizukami2, Wouter J. M. Knoben1, Jefferson S.</li> <li>Wong3, Alain Pietroniro4</li>                                                                                                                                                                                                                                                                                                                                                                                                                                                                                                                                                                                                                                                                                             |                               |

[revised manuscript text omitted]